# Design and characterization of a protein fold switching network

Biao Ruan[1,6], Yanan He[2,6], Yingwei Chen[1,6], Eun Jung Choi[1], Yihong Chen[2], Dana Motabar [1,3], Tsega Solomon[2,4], Richard Simmerman[1], Thomas Kauffman[2,4], D. Travis Gallagher[2,5], John Orban [2,4] ✉ & Philip N. Bryan [1,2] ✉

To better understand how amino acid sequence encodes protein structure, we engineered mutational pathways that connect three common folds (3α, β−grasp, and α/β−plait). The structures of proteins at high sequence-identity intersections in the pathways (nodes) were determined using NMR spectroscopy and analyzed for stability and function. To generate nodes, the amino acid sequence encoding a smaller fold is embedded in the structure of an ~50% larger fold and a new sequence compatible with two sets of native interactions is designed. This generates protein pairs with a 3α or β−grasp fold in the smaller form but an α/β−plait fold in the larger form. Further, embedding smaller antagonistic folds creates critical states in the larger folds such that single amino acid substitutions can switch both their fold and function. The results help explain the underlying ambiguity in the protein folding code and show that new protein structures can evolve via abrupt fold switching.

There have been remarkable advances recently in the ability to predict the tertiary structure of a protein from its primary amino acid sequence[1,2] as well as to design amino acid sequences that encode stable, unique protein structures[3]. It is also well-established, however, that some proteins have a propensity for two completely different, but well-ordered, conformations[4–12]. Better insight into the ambiguity of the protein folding code would lead to a better understanding of how proteins evolve, how mutation is related to disease, and how function can be annotated to sequences of unknown structure[13–27]. If the protein folding code were truly understood, it would be possible both to predict and design proteins that undergo profound switches in conformation. There has been significant progress in understanding natural proteins that switch folds[11] and predicting natural fold-switching proteins from amino acid sequence data[25]. Designing proteins at the interface between different folds has been possible[7,28–30] but still presents a formidable challenge. It has been particularly challenging to design monomeric proteins that switch fold without a change in quaternary structure, and a better understanding is needed about how a very limited subset of intra-protein interactions can tip the balance from one fold and function to another[29,31,32].

Our goal here was to engineer monomeric proteins that are in a critical state between two distinct folds. To do this we chose three well-studied protein folds and designed a series of sequences such that each sequence is compatible with two sets of native interactions. Two of these folds are from *Streptococcal* Protein G which contains two types of domains that bind to serum proteins in blood: the $G_A$ domain binds to human serum albumin (HSA)[33,34] and the $G_B$ domain binds to the constant (Fc) region of IgG[35,36]. The third protein is S6, a component of the 30S ribosomal subunit of *Thermus thermophilus*[37–41]. For simplicity, the S6 fold is referred to as an S-fold, the $G_A$ fold as an A-fold, and the $G_B$ fold as a B-fold. These proteins share no significant sequence homology and are representative of three of the ten most common folds: the S-fold is a thioredoxin-like α/β plait; the A-fold is a homeodomain-like 3α-helix bundle; and the B-fold is a ubiquitin-like β grasp[42].

Figure 1 depicts a network of high-identity sequence intersections (nodes) that connect the three folds. The arrows in Fig. 1 show a

[1]Potomac Affinity Proteins, 11305 Dunleith Pl, North Potomac, MD 20878, USA. [2]Institute for Bioscience and Biotechnology Research, University of Maryland, 9600 Gudelsky Drive, Rockville, MD 20850, USA. [3]Department of Bioengineering, University of Maryland, College Park, MD 20742, USA. [4]Department of Chemistry and Biochemistry, University of Maryland, College Park, MD 20742, USA. [5]National Institute of Standards and Technology and the University of Maryland, 9600 Gudelsky Drive, Rockville, MD 20850, USA. [6]These authors contributed equally: Biao Ruan, Yanan He, Yingwei Chen. ✉e-mail: jorban@umd.edu; pbryan@potomac-affinity-proteins.com

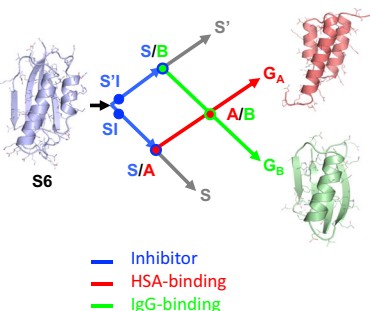

**Fig. 1 | Overview of engineered nodes in the S6, G_A, and G_B networks.** S6 is the origin sequence in the engineering process. SI and S'I are separate nodes and are loop-size variants of the S-fold, both having protease inhibitor functions. The SI branch of the mutational path leads to a node with the A-fold and HSA binding function. The S'I branch of the path leads to a node with the B-fold and IgG binding function. The S/A node (blue and red circles) includes proteins $S_{a1}$, $S_{a2}$, $A_1$, and $A_2$. The S/B node (blue and green circle) includes proteins $S_{b3}$, $S_{b4}$, $S_{b5}$, $B_3$, and $B_4$. The A and B paths themselves intersect at an A/B node (green and red circles) at which A- and B-folds are nearly iso-energetic and bifunctional. The S and S' branches continue and connect with many other natural sequences in the α/β plait super-fold family.

network originating with the natural S6 sequence. Circles represent nodes in the network at which structural and/or functional switches occur. The SI and S'I nodes are branch points and lead down diverging sequence pathways, one leading to a node with the A-fold (S/A) and one to a node with the B-fold (S/B). Intersecting mutational pathways lead from S/A to the native $G_A$ protein and S/B to the native $G_B$ protein. At this intersection (A/B), an A-fold switches to a B-fold.

Proteins around the A/B node have been extensively characterized in our earlier work[29,31,32]. Here we determine that both $G_A$ and $G_B$ can switch into a third fold (α/β−plait) and show that these three folds and four functions (HSA-binding, IgG-binding, protease inhibition, and RNA-binding) can be connected in a network that avoids unfolded and functionless states. We describe how these nodes were engineered, determine key structures using NMR spectroscopy, and analyze stability and binding function. The ability to design and characterize nodes connecting three common small folds suggests that fold switching may be an intrinsic feature of the protein folding code and is important in the evolution of protein structure and function.

# Results

## Designing a functional switch from ribosomal protein to protease inhibitor

The S6 ribosomal protein is structurally homologous to subtilisin protease inhibitors known as prodomains (Fig. 2a, b)[43,44]. Prodomain-type inhibitors have two binding surfaces with the protease. One surface comprises the last nine C-terminal amino acids of the inhibitor which bind in the substrate binding cleft of the protease (Fig. 2b). A second, more dynamic surface is formed between two subtilisin helices and the large surface of the β−sheet in the α/β-plait topology of the inhibitor (Fig. 2b)[45–47]. As a result, the S6 protein could be converted into a subtilisin inhibitor protein of the same overall fold (denoted SI) by replacing its nine C-terminal amino acids with residues optimized to bind in the substrate binding cleft of subtilisin. This replacement results in new contacts between the SI β−sheet and the subtilisin surface helices (Fig. 2b).

The SI-protein is 99 amino acids in length and has a 10 residue loop between β2 and β3. However, there are many natural variations in the length of loops in the conserved α/β-plait topology[48]. Therefore, we also engineered a 91 amino acid version of the S-fold (denoted S'I), which resembles the topology of natural prodomain inhibitors (Supplementary Fig. 1). Specifically, the S'I inhibitor has a longer loop connecting β1 to α1 and a shorter turn connecting β2 to β3 (Fig. 2b).

The SI and S'I proteins were expressed and purified by binding to a protease column[49]. The CD spectra were compared to the native S6 protein (Supplementary Fig. 1). Inhibition constants ($K_I$) were measured using an engineered RAS-specific subtilisin protease and the peptide substrate QEEYSAM-AMC[49]. SI and S'I inhibit the RAS-specific protease with $K_I$ values of 200 and 60 nM, respectively (Supplementary Table 1). The details of the competitive inhibition assay are described in the "Methods" section. The results demonstrate that a ribosomal protein can be converted into a protease inhibitor with minor modification (and without a fold switch). In addition, however, the SI and S'I proteins also facilitated engineering subsequent switches to new folds and functions by linking each of the S-, A-, and B- folds to easily measured binding functions: protease inhibition (S or S'-fold); HSA-binding (A-fold, Fig. 2e)[50]; and IgG binding (B-fold, Fig. 2f)[51].

## Designing fold switches

In previous work, we created sequences that populate both A- and B-folds by threading the A-sequence through the B-fold, finding a promising alignment, and then using phage-display selection to reconcile one sequence to both folds[29,52,53]. Here the approach is conceptually similar, except that we use Rosetta[54] as a computational design tool to test compatible mutations rather than phage display. The design process is as follows:

i. Thread the A- or B- sequence through both SI and S'I-fold types.
ii. Identify alignments that minimize the number of catastrophic interactions.
iii. Design mutations to resolve unfavorable interactions in clusters of 4−6 amino acids using Pymol[55] and energy minimize using Rosetta-Relax[54].
iv. Optimize protein stability in the S-fold by computationally mutating amino acids at non-overlapping positions. Repeat energy minimization and evaluation with Rosetta-Relax.
v. To reduce uncertainties involved in computational design, conserve original amino acids whenever possible.

There is no reason to assume that this method is optimal. We are just applying a practicable scheme for engineering sequences compatible with two sets of native interactions and then evaluating structure, stability, and function. Initial designs were refined based on structural analysis with NMR, thermodynamic analysis of unfolding, and functional analysis using binding assays, as described below. All designed proteins were expressed in *E. coli* and purified to homogeneity as described in the "Methods" section.

## Designing a switch from α/β-plait protease inhibitor to 3α HSA-binding protein

Alignment of the 56 amino acid HSA-binding, A-fold with the 99 amino acid SI-fold and subsequent mutation to resolve catastrophic interactions produced low-energy switch candidates denoted $S_{a1}$ and $A_1$. The exact sequence of $A_1$ is embedded in $S_{a1}$ at positions 11–66 such that the α1 helices are structurally aligned (Fig. 3a, Supplementary Fig. 2A). Their final computational models were generated by Rosetta using the Relax application. The Relax protocol searches the local conformational space around an experimentally determined structure and is used only to evaluate whether the designed mutations have favorable native interactions within that limited conformational space. The designed models of $S_{a1}$ and $A_1$ are very similar in energy compared to the respective relaxed native structures (Supplementary Fig. 3 and Source data files).

## Structural analysis of $A_1$ and $S_{a1}$

Overall, the 3α-helical bundle topology of $A_1$ is very similar to the $G_A$ parent structure from which it was derived[56]. The sequence-specific chemical shift assignments for $A_1$ (Fig. 3b) were utilized to calculate a 3D structure with CS-Rosetta (Fig. 3c, Table 1). Our previous studies

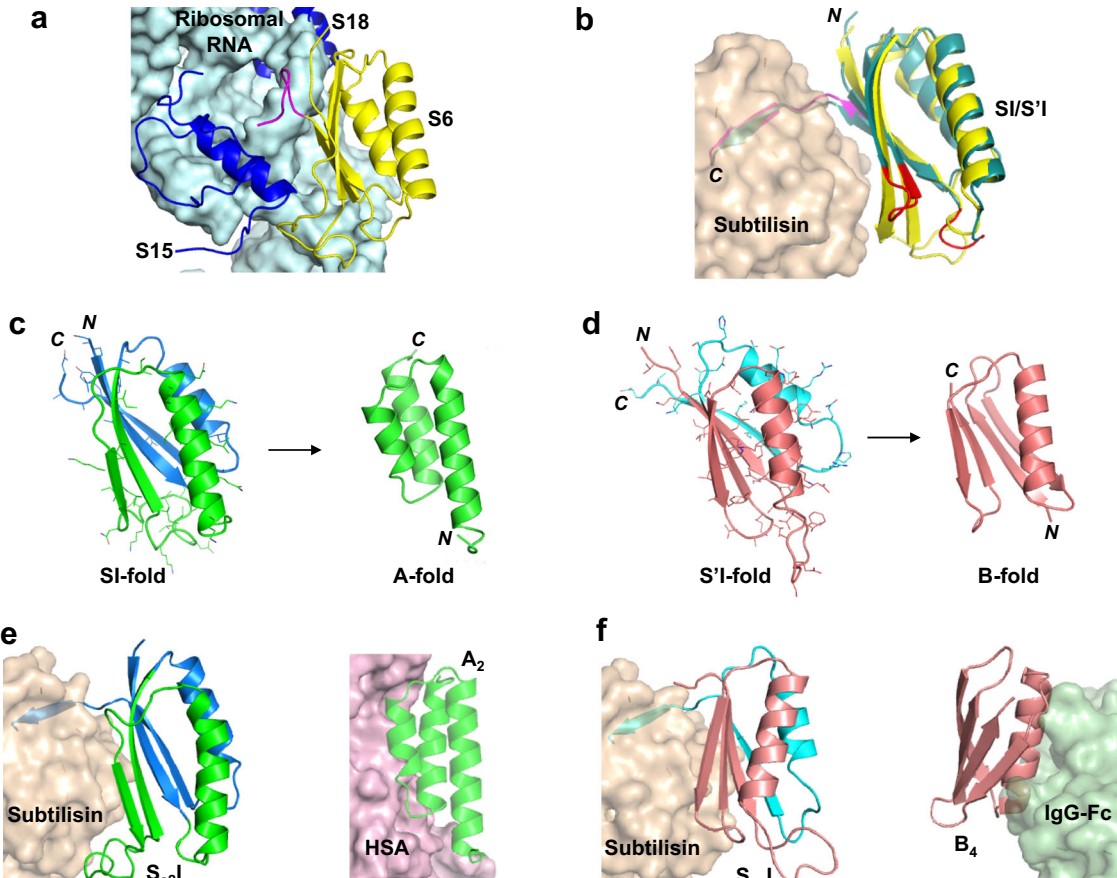

**Fig. 2 | Summary of switches in structure and function. a** Structure of the S6 protein (yellow), RNA (light blue), and S15 and S18 proteins (blue) in the 30S ribosome (PDB 1FKA [https://doi.org/10.2210/pdb1FKA/pdb], ref. [43]). C-terminal amino acids of S6 are in magenta. **b** Subtilisin (wheat) is shown in complex with a model of the SI-inhibitor (yellow). The C-terminal nine amino acids of SI are shown in magenta. These positions were mutated in native S6 to generate affinity for subtilisin. The S′I-inhibitor (teal) is also shown with the altered loops in red. The subtilisin used in the modeling was the engineered RAS-specific protease. **c** The $S_{a1}$ protein (blue and green) was generated from SI by mutating the 45 positions (mutant side chains shown with sticks). Deletion of C-terminal amino acids (blue) switches $S_{a1}$ into the A-fold (green). **d** The $S_{b3}$ protein (rose and cyan) was generated from S′I by mutating the 67 positions (mutant side chains shown with sticks). Deletion of C-terminal amino acids (cyan) or point mutation will switch $S_{b3}$ into a B-fold (rose). **e** Model of $S_{a2}$I (green and blue) bound to subtilisin (wheat). Model of $A_2$ (based on $A_1$ structure) bound to HSA (violet). The HSA complex used PDB 2VDB (ref. [50]) as the template. **f** Model of $S_{b3}$ (rose and cyan) bound to subtilisin (wheat). Model of $B_4$ (rose) bound to Fc (mint). The Fc complex used PDB 1FCC (ref. [51]) as the template. The subtilisin used in the modeling and inhibition measurements was the engineered RAS-specific protease PDB 6UAO (ref. [49]).

indicated close correspondence of CS-Rosetta and de novo structures for A- and B-folds with high sequence identity[57]. The N-terminal residues 1–4 and the C-terminal residues 53–56 are disordered in the structure, consistent with {¹H}-¹⁵N steady-state heteronuclear NOE data (Fig. 3e). Likewise, $S_{a1}$ has the same overall βαββαβ-topology as the parent S6 structure (Fig. 3d, Table 2). The backbone chemical shifts (Fig. 3b) were used in combination with main chain inter-proton NOEs (Supplementary Fig. 4) to determine a three-dimensional structure utilizing CS-Rosetta (PDB 7MN1). The conformational ensemble shows well-defined elements of secondary structure at residues 2–10 (β1), 16–32 (α1), 40–44 (β2), 59–67 (β3), 73–81 (α2), and 86–92 (β4). The principal difference from the native structure is that the β2-strand is seven amino acids shorter in $S_{a1}$ than in S6. Heteronuclear NOE data show overall consistency with the structure, indicating that the long loop between the β2- and β3-strands from residues 45–58 is more flexible than other internal regions of the polypeptide chain (Fig. 3e).

### Comparison of $A_1$ and $S_{a1}$ structures

Although the 56 amino acid sequence of $A_1$ is 100% identical to residues 11–66 of $S_{a1}$, a significant fraction of the backbone undergoes changes between the two structures. Most notably, while the α1 helices in both $A_1$ and $S_{a1}$ are similar in length, the regions corresponding to

the α2 and α3 helices of $A_1$ form the β2 and β3 strands of $S_{a1}$ (Fig. 4a). Core amino acids in the α1-helix of $A_1$ correspond with residues that also contribute to the core of $S_{a1}$. However, the α1-helix in $S_{a1}$ contacts an almost entirely different set of residues (Fig. 4b). For example, amino acids L51, Y53, and I55 in the C-terminal tail of $A_1$ do not have extensive contact with α1 but the corresponding residues in $S_{a1}$ (L61, Y63, and I65) form close core interactions with α1 as part of the β3-strand. Most of the other core residues contacting the α1-helix of $S_{a1}$ are outside the 56 amino acid region coding for the $A_1$ fold. These include F4, V6, I8, and L10 from the β1-strand; A67 from the β3-strand; V72, L75, and L79 from the α2-helix; and V85 from the loop between the α2-helix and the β4-strand. Two additional residues, V88 and V90 (β4) also contribute significantly to the core but do not contact α1. Thus, except for the original topological alignment of the α1-helices, the cores of the 3α and α/β-plait folds are largely non-overlapping. In total, approximately half of the residues participating in the $S_{a1}$ core are not present in the $A_1$ sequence.

### Energetics of unfolding for $A_1$/$S_{a1}$

Far-UV CD spectra were measured for $S_{a1}$ and $A_1$ and their thermal unfolding profiles were determined by measuring ellipticity at 222 nm versus temperature (Fig. 5 and Supplementary Fig. 5). $S_{a1}$ has a

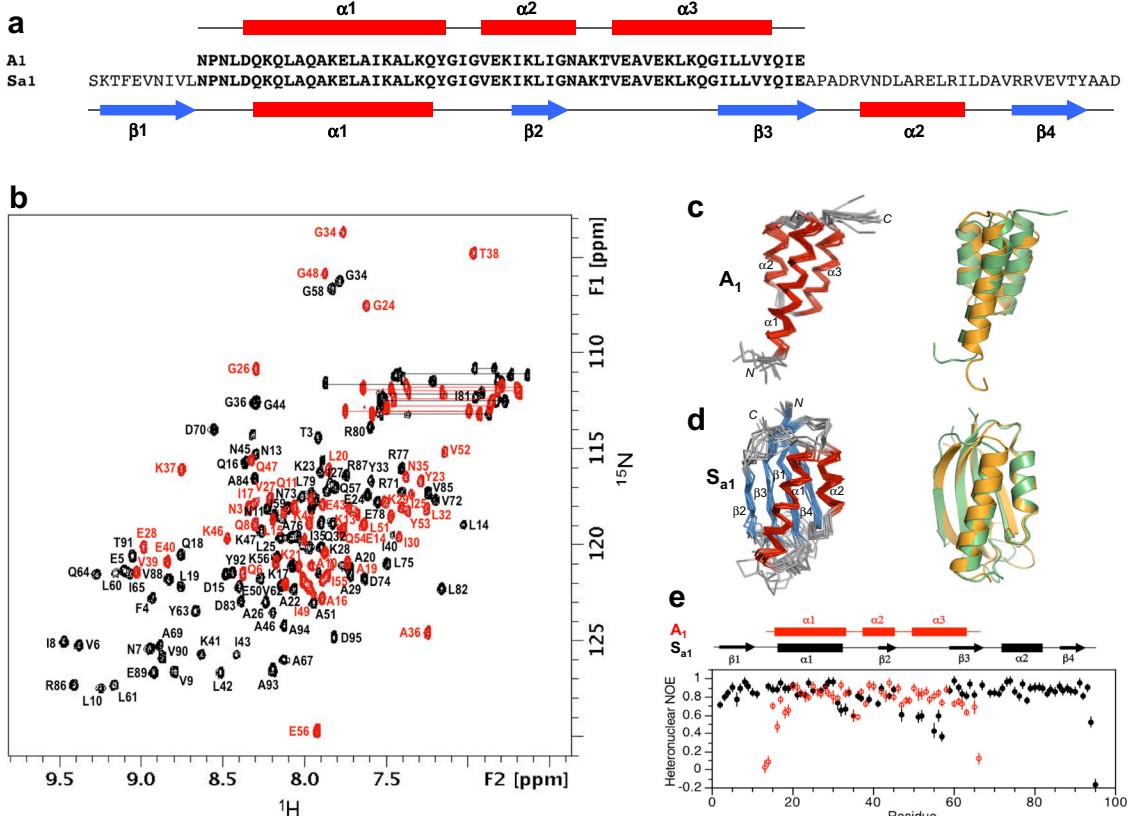

**Fig. 3 | Structure and dynamics of A₁ and S_{a1}. a** Sequence alignment of A₁ and S_{a1}, which are 100% identical over the 56 amino acid A-region. **b** Overlaid two-dimensional ¹H-¹⁵N HSQC spectra of S_{a1} (black) and A₁ (red) with backbone amide assignments. Spectra were recorded at 25 and 5 °C, respectively. **c** Ensemble of 10 lowest energy CS-Rosetta structures for A₁ (left panel). Superposition of the A₁ structure (green) with the parent G_A fold (orange) (right panel). **d** Ensemble of 10 lowest energy CS-Rosetta structures for S_{a1} (left panel). Superposition of S_{a1} (green) with the parent S6 fold (orange) (right panel). **e** Backbone dynamics in designed proteins. Plot of {¹H}-¹⁵N steady state heteronuclear NOE values at 600 MHz versus residue for A₁ (red) and for S_{a1} (black). Each set of heteronuclear NOEs was obtained from a single experiment. Errors were estimated based on the measured background noise level.

$T_M$ of ~100 °C and an estimated $\Delta G_{folding}$ of −5.3 kcal/mol at 25 °C (Fig. 5b, Supplementary Table 1)[58]. The $\Delta G_{folding}$ of the parent S6 is −8.5 kcal/mol[40]. The Rosetta energy of the S_{a1} design is similar to that of the native sequence (Supplementary Fig. 3). A₁ has a $T_M$ of 65 °C and a $\Delta G_{folding}$ = −4.0 kcal/mol at 25 °C[58] (Fig. 5a, Supplementary Table 1). The $\Delta G_{folding}$ of the parent $G_A$ is −5.6 kcal/mol[59,60]. The Rosetta energy of the A₁ design is slightly more favorable than for the native sequence (Supplementary Fig. 3).

## HSA binding
Initial engineering of the fold switch was carried out without consideration of preserving function. As a result, A₁ does not have detectable HSA binding affinity because two amino acids in the binding interface were mutated. Significant HSA-binding is recovered, however, when the surface mutations, E28Y and K29Y, are made in A₁ (denoted A₂). These mutations do not appear to affect the structure of A₁ (Supplementary Fig. 5) but result in HSA binding of $K_D ≤ 1 \mu M$ (Supplementary Table 1). This was determined by measuring binding to immobilized HSA as described in the "Methods" section.

## Protease inhibition
S_{a1} does not bind protease because C-terminal amino acids were not preserved in its design. It can be converted into a protease inhibitor, however, by replacing its three C-terminal amino acids (AAD) with DKLYRAL (denoted S_{a1}I). A version of S_{a1}I was also made that contains the exact 56 amino acid A₂ sequence by making E38Y, K39Y mutations (denoted S_{a2}I). S_{a1}, S_{a1}I, and S_{a2}I are

similar in structure by CD analysis (Supplementary Fig. 5). The inhibition constant of S_{a2}I with the engineered subtilisin was determined to be 50 nM as described in the "Methods" section (Supplementary Table 1). Thus, a stable A-fold with HSA-binding function can be embedded within a 99 amino acid S-fold with protease inhibitor function (Fig. 2c, e). It should be noted that all HSA contact amino acids are preserved in both the A₂ and S_{a2}I sequences, but the three-dimensional topology necessary to form the HSA contact surface occurs only in the A-fold[50]. Nevertheless, S_{a2}I was observed to bind weakly to HSA ($K_D$ ~ 100 μM, Supplementary Table 1). This weak affinity suggests that some S_{a2}I molecules may populate the 3α fold even though the α/β-plait fold strongly predominates.

## Designing a switch from α/β-plait protease inhibitor to β−grasp IgG-binding protein
In designing an S- to B-fold switch, we used two topological alignments. The first was between SI- and B-folds, where the β1 strands of each fold were aligned (Supplementary Figs. 2B and 6A). The second alignment was between S′I- and B-folds, where the long loop between β2 and β3 in SI was shortened in S′I to be more consistent with natural protease inhibitors. In this scheme, the α1β3β4 topology of the B-fold was aligned with the α1β2β3 topology of the S′I-fold (Fig. 6a, Supplementary Fig. 2C).

## Design and characterization of B₁, S_{b1}, B₂, and S_{b2}
In the first approach, alignment of the β1-strands of the B-fold and the S-fold and subsequent mutation to resolve catastrophic interactions

**Table 1 | Structure statistics for $A_1$, $B_1$, and $B_4$**

|  | $A_1$ | $B_1$ | $B_4$ |
|---|---|---|---|
| **A. Experimental chemical shift inputs** | | | |
| $^{13}C\alpha$ | 55 | 56 | 56 |
| $^{13}C\beta$ | 51 | 50 | 53 |
| $^{13}CO$ | 54 | 55 | 53 |
| $^{15}N$ | 54 | 55 | 54 |
| $^{1}H_N$ | 54 | 55 | 54 |
| $^{1}H\alpha$ | – | 55 | – |
| **B. RMSDs to the mean structure (Å)** | | | |
| Over all residues | | | |
| Backbone atoms | 1.73 ± 0.47 | 0.86 ± 0.22 | 0.85 ± 0.34 |
| Heavy atoms | 2.27 ± 0.53 | 1.37 ± 0.29 | 1.43 ± 0.45 |
| Secondary structures[a] | | | |
| Backbone atoms | 0.90 ± 0.29 | 0.67 ± 0.22 | 0.62 ± 0.25 |
| Heavy atoms | 1.55 ± 0.42 | 1.12 ± 0.25 | 1.33 ± 0.41 |
| **C. Measures of structure quality (%)** | | | |
| Ramachandran distribution | | | |
| Most favored | 98.78 ± 1.64 | 92.31 ± 3.35 | 92.36 ± 2.71 |
| Additionally allowed | 1.22 ± 1.64 | 7.69 ± 3.36 | 7.64 ± 2.71 |
| Generously allowed | 0.00 ± 0.00 | 0.00 ± 0.00 | 0.00 ± 0.00 |
| Disallowed | 0.00 ± 0.00 | 0.00 ± 0.00 | 0.00 ± 0.00 |
| **D. Backbone RMSDs to the parent structure[b] (Å)** | | | |
| Over all residues | 2.48 | 0.62 | 0.64 |
| Secondary structures | 1.21 | 0.57 | 0.49 |
| **E. PDB/BMRB codes** | | | |
| PDBDEV | 00000083 | 00000084 | 00000085 |
| BMRB | 50,907 | 50,910 | 50,909 |

[a]The secondary elements used were as follows: $A_1$, residues 5–23, 27–35, 39–53; $B_1$, residues 2–8, 13–19, 23–37, 43–46, 51–55; $B_4$, residues: 2–8, 13–19, 23–37, 42–46, 51–55.
[b]The parent structure for $A_1$ is $G_A$ (PDB 2FS1). The parent structure for $B_1$ and $B_4$ is $G_B$ (PDB 1PGA). RMSDs were calculated by superimposing the mean structure from the NMR ensemble with either the mean structure (in the case of 2FS1) or the X-ray structure (in the case of 1PGA) for the parent.

**Table 2 | Structure statistics for $S_{a1}$, $S_{b1}$, $S_{b2}$, and $S_{b3}$**

|  | $S_{a1}$ | $S_{b1}$ | $S_{b2}$ | $S_{b3}$ |
|---|---|---|---|---|
| **A. Experimental restraint inputs** | | | | |
| NOE restraints | | | | |
| Sequential ($i$–$j$ = 1) | 89 | – | 35 | 40 |
| Medium range (1< $i$–$j$ ≤ 5) | 35 | – | 31 | 10 |
| Long range ($i$–$j$ > 5) | 92 | – | 67 | 66 |
| Hydrogen bond restraints | 88 | – | 82 | 83 |
| TALOS dihedral angle restraints | – | – | – | 91 |
| Total NOE restraint inputs | 304 | | 215 | 290 |
| PRE restraints | | 41 | | |
| **B. Experimental chemical shift inputs** | | | | |
| $^{13}C\alpha$ | 88 | 79 | 83 | – |
| $^{13}C\beta$ | 86 | 70 | 76 | – |
| $^{13}CO$ | 69 | 67 | 72 | – |
| $^{15}N$ | 88 | 69 | 76 | – |
| $^{1}H_N$ | 88 | 69 | 76 | – |
| $^{1}H\alpha$ | 76 | 45 | 61 | – |
| **C. RMSDs to the mean structure (Å)** | | | | |
| Over all residues[a] | | | | |
| Backbone atoms | 2.77 ± 0.82 | 5.47 ± 1.86 | 2.34 ± 0.60 | 2.46 ± 0.64 |
| Heavy atoms | 3.51 ± 0.85 | 6.32 ± 1.86 | 3.00 ± 0.62 | 3.29 ± 0.61 |
| Secondary structures[b] | | | | |
| Backbone atoms | 1.07 ± 0.23 | 3.79 ± 1.50 | 1.08 ± 0.24 | 0.68 ± 0.14 |
|  | | (0.71 ± 0.23)[c] | | |
| Heavy atoms | 1.84 ± 0.37 | 4.37 ± 1.42 | 1.78 ± 0.32 | 1.42 ± 0.25 |
|  | | (1.24 ± 0.30)[c] | | |
| **D. Measures of structure quality (%)** | | | | |
| Ramachandran distribution | | | | |
| Most favored | 86.46 ± 4.07 | 92.35 ± 2.38 | 92.19 ± 1.83 | 86.30 ± 2.38 |
| Additionally allowed | 13.54 ± 4.07 | 7.54 ± 2.60 | 7.81 ± 1.83 | 10.52 ± 2.68 |
| Generously allowed | 0.00 ± 0.00 | 0.00 ± 0.00 | 0.00 ± 0.00 | 1.47 ± 1.54 |
| Disallowed | 0.00 ± 0.00 | 0.12 ± 0.38 | 0.00 ± 0.00 | 1.96 ± 1.10 |
| **E. Backbone RMSDs to the parent structure (Å)[d]** | | | | |
| Over all residues | 2.37 | 0.49 | 6.16 | 11.67 |
| Secondary structures | 0.88 | 0.41 | 3.39 | 3.20 |
| **F. PDB/BMRB codes** | | | | |
| PDB | 7MN1 | 7MQ4 | 7MN2 | 7MP7 |
| BMRB | 30,901 | 30,905 | 30,902 | 30,904 |

[a]Over all residues used as follows: $S_{a1}$, 1–95, $S_{b1}$, 4–85, $S_{b2}$, 1–93, $S_{b3}$, 1–87.
[b]The secondary elements used are as follows: $S_{a1}$, residues 2–10, 16–32, 40–44, 59–67, 72–81, 86–92; $S_{b1}$, residues 5–12, 17–24, 27–41, 46–50, 55–59, 73–83; $S_{b2}$, residues 2–9, 23–32, 43–48, 59–65, 71–80, 85–91; $S_{b3}$, residues 4–10, 24–37, 40–46, 51–57, 62–71, 79–85.
[c]RMSDs for $S_{b1}$ minus the putative α2 region: residues 5–12, 17–24, 27–41, 46–50, 55–59.
[d]The parent structure for $S_{a1}$, $S_{b2}$, and $S_{b3}$ is PDB 1RIS. The parent structure for $S_{b1}$ is PDB 1PGA. In this case, the structure alignment is over the 56 amino acid B-region of $S_{b1}$.

produced low-energy switch candidates denoted $B_1$ and $S_{b1}$. The exact sequence of $B_1$ is embedded in $S_{b1}$ at positions 4–59 (Supplementary Fig. 6A). The computational models of $B_1$ and $S_{b1}$ show relatively small increases in energy compared to the corresponding relaxed native structures (Supplementary Fig. 3). The NMR structure of $B_1$ displayed a ββαββ topology identical to that of the parent B-fold, with a backbone RMSD of ~0.6 Å (Supplementary Fig. 6B, C). The topology of $S_{b1}$ is not the same as the parent S6 structure, however, and instead has a fold similar to that of $B_1$ (Supplementary Figs. 6B, D, and 7, PDB 7MQ4). Introducing 13 mutations into $S_{b1}$ generated a protein denoted $S_{b2}$ (Supplementary Fig. 8). $S_{b2}$ contains four β-strands and two α-helices and has the general features of the parent S-fold (Supplementary Fig. 9, PDB 7MN2). The 56 amino acid version of $S_{b2}$ (denoted $B_2$) has a significantly higher Rosetta energy than $B_1$, however, and is presumably unfolded (Supplementary Fig. 3). Thus, neither the $B_1$/$S_{b1}$ nor $B_2$/$S_{b2}$ protein pairs resulted in high identity sequences with different folds. Nonetheless, $B_1$ is 80% identical to the corresponding embedded region in the S-folded protein $S_{b2}$ (Supplementary Fig. 9A). The structures of $B_1$, $S_{b1}$, and $S_{b2}$ are described further in the Supplement and Tables 1 and 2.

### Design of $S_{b3}$ and $B_3$
To improve the design of the S-to-B switch we aligned the B-fold with the S′ inhibitor fold and chose an alignment that creates a topological match between α1β3β4 in B and α1β2β3 in S′ (Supplementary Fig. 2C). Mutation to resolve deleterious interactions

in this alignment produced low-energy switch candidates denoted $B_3$ and $S_{b3}$ (Supplementary Fig. 10). The exact sequence of $B_3$ is embedded in $S_{b3}$ at positions 1–56. The energy of the computational model for $S_{b3}$ is slightly more favorable than the relaxed native structure. The designed model of $B_3$ shows relatively small

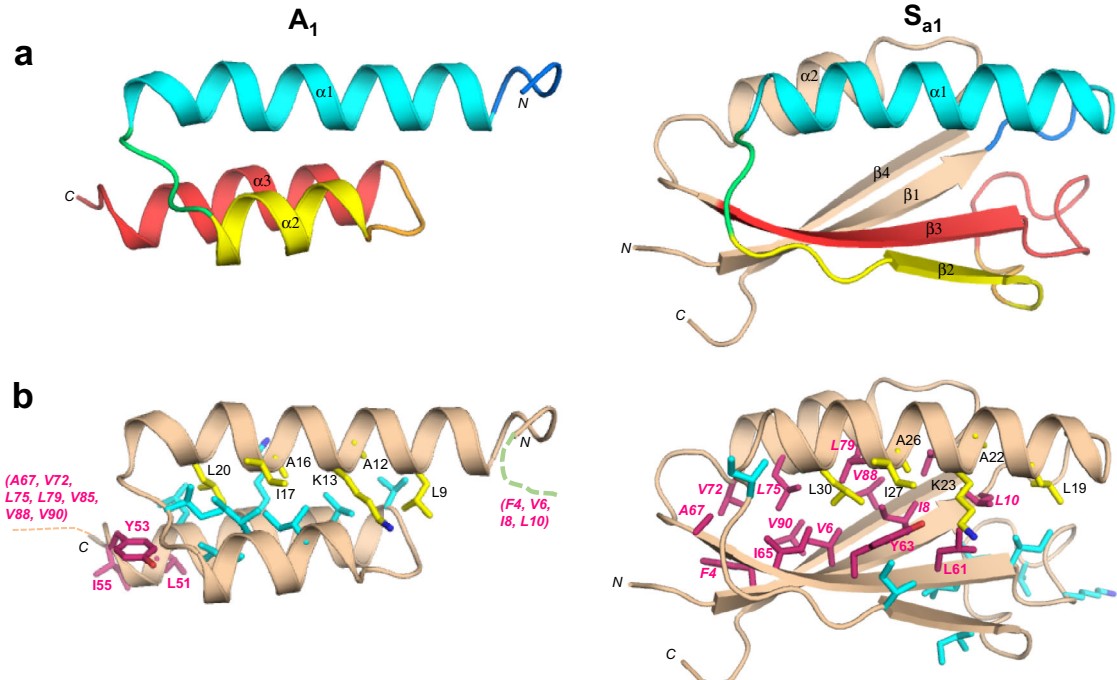

**Fig. 4 | Structural differences between the 100% sequence identical regions of A₁ and S_a1. a** Main chain comparisons. (Left panel) CS-Rosetta structure of A₁ with color coding for secondary structured elements. (Right panel) Corresponding color-coded regions mapped onto the CS-Rosetta structure of S_a1, illustrating changes in backbone conformation. Regions outside the 56 amino acid sequence of A₁ are shown in wheat. **b** Side chain comparisons. (Left panel) Residues contributing to the core of A₁ from the α1-helix (yellow), and from other regions (cyan). The non-α1 core residues from S_a1 (pink) do not overlap with the A₁ core (see text for further details). (Right panel) Residues contributing to the core of S_a1 from the α1-helix (yellow), and most of the other participating core residues (pink). The non-α1 core residues from A₁ are also shown (cyan), highlighting the low degree of overlap.

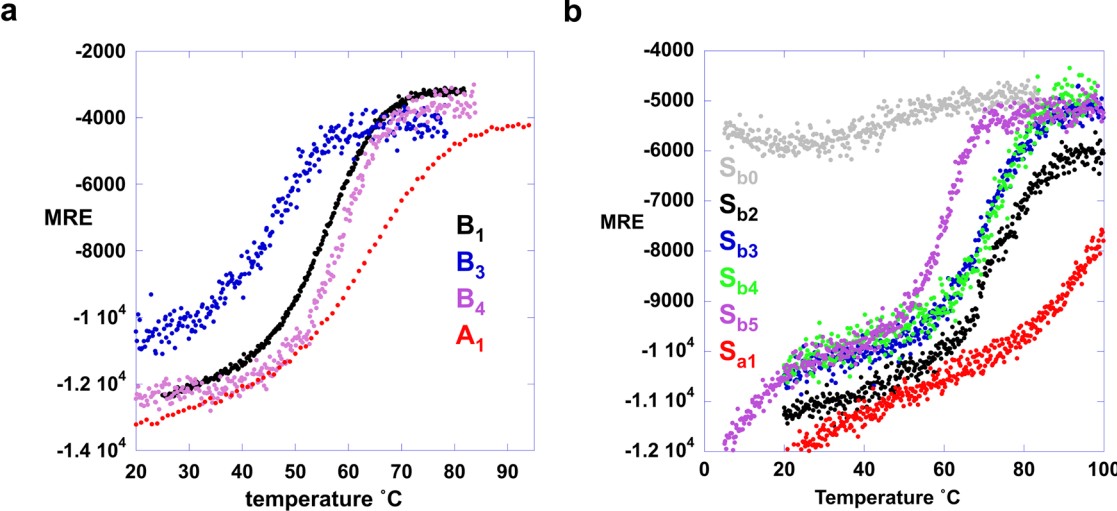

**Fig. 5 | CD melting curves. a** Ellipticity at 222 nm plot versus temperature for A- and B- variations. **b** Ellipticity at 222 nm plot versus temperature for S-variations. S_b0 is a low-stability variant (F7V) of S_b1 used to measure the temperature dependence of the unfolded state. Source data are provided as a Source Data file.

increases in energy compared to the relaxed native structure (Supplementary Fig. 3).

## Structural analysis of S_b3 and B₃

NMR-based structure determination indicated that S_b3 has a βαββαβ secondary structure and an S-fold topology (Fig. 6a, b, d, PDB 7MP7). Ordered regions correspond with residues 4–10 (β1), 24–37 (α1), 42–46 (β2), 51–56 (β3), 62–70 (α2), and 79–85 (β4). Comparison of S_b3 with the parent S-fold indicates that the β1/α2/β4 portion of the fold is similar in both. In contrast, the β1–α1 loop is longer in S_b3 (13 residues) than in the parent S-fold (5 residues), while α1, β2, the β2–β3 loop, and

β3 are all shorter than in the parent (Fig. 6d). Consistent with the S_b3 structure, the 13 amino acid β1-α1 loop is highly flexible (Fig. 6e). We also expressed and purified a truncated protein corresponding to the embedded B-fold, the 56 amino acid version of S_b3 (denoted B₃). The 2D ¹H–¹⁵N HSQC spectrum of B₃ at 5 °C and low concentrations (<20 μM) was consistent with a predominant, monomeric B-fold (Supplementary Fig. 11) but showed significant exchange broadening at 25 °C, indicative of low stability (see below). Presumably, the low stability is due to the less favorable packing of Y5 in the core of the B-fold compared with a smaller aliphatic leucine. However, additional, putatively oligomeric, species were also present for which relative

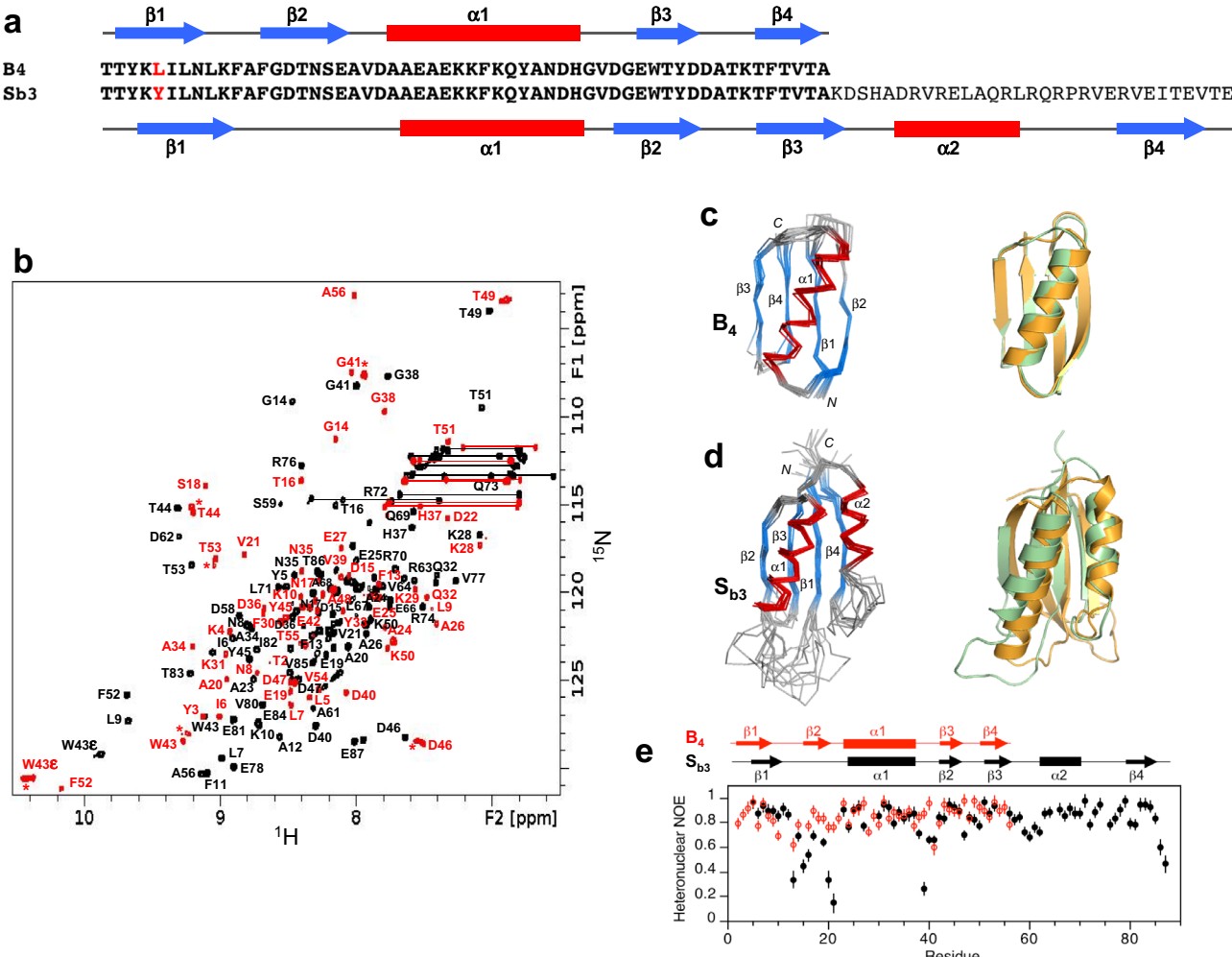

**Fig. 6 | Structure and dynamics of S<sub>b3</sub> and B<sub>4</sub>. a** Sequence alignment of B<sub>4</sub> and S<sub>b3</sub>, differing by one residue (L5Y) over the 56 amino acid B-region. **b** Overlaid two-dimensional ¹H-¹⁵N HSQC spectra of S<sub>b3</sub> (black) and B<sub>4</sub> (red) with backbone amide assignments. Spectra were recorded at 25 °C. The A56 peak is an aliased signal. Peaks labeled with an asterisk decrease in relative intensity as the B<sub>4</sub> concentration is lowered, indicating the presence of a weakly associated putative dimer in addition to monomer. **c** Ensemble of 10 lowest energy CS-Rosetta structures for B<sub>4</sub> (left panel). Superposition of the B<sub>4</sub> structure (green) with the parent G<sub>B</sub> fold (orange) (right panel). **d** Ensemble of 10 lowest energy CS-Rosetta structures for S<sub>b3</sub> (left panel). Superposition of S<sub>b3</sub> (green) with the parent S6 fold (orange) (right panel). **e** Plot of {¹H}-¹⁵N steady-state heteronuclear NOE values at 600 MHz versus residue for B<sub>4</sub> (red) and S<sub>b3</sub> (black). Each set of heteronuclear NOEs was obtained from a single experiment. Errors were estimated based on the measured background noise level.

peak intensities increased with increasing protein concentration. Due to its relatively low stability and sample heterogeneity, B₃ was not analyzed further structurally.

**Design and analysis of point mutations that switch the fold of S<sub>b3</sub>**

We used the NMR structure of S<sub>b3</sub> to design a point mutation, tyrosine 5 to leucine (Y5L), that would stabilize the embedded B-fold without compromising native contacts in the S-fold (Supplementary Figure 10). This mutant was therefore expected to shift the population to the B-fold. Two mutants were prepared, a Y5L mutant of S<sub>b3</sub> (denoted S<sub>b4</sub>) and a Y5L mutant of B₃ (denoted B<sub>4</sub>). B<sub>4</sub>, is indeed more stable than B₃ (Fig. 5a, Supplementary Table 1). Assignment and structure determination of B<sub>4</sub> showed its topology to be identical to the parent B-fold (Fig. 6b, c). At concentrations above 100 μM, B<sub>4</sub> displayed a tendency for weak self-association similar to that seen for B₃. For S<sub>b4</sub>, the HSQC spectrum exhibited approximately twice the number of amide cross-peaks relative to S<sub>b3</sub> (Fig. 7a), suggesting that S- and B-states were populated simultaneously. This was confirmed by the NMR assignment and also a comparison of the HSQC spectra for S<sub>b4</sub>, B<sub>4</sub>, and S<sub>b3</sub>. A significant fraction of the S<sub>b4</sub> backbone amide signals (~50 peaks) closely matched those of B<sub>4</sub>, indicating the presence of a B-state (Supplementary Fig. 12A–C). The

close matching of these peaks is presumably because residues 1–56 in the B-state of S<sub>b4</sub> are identical in sequence to B<sub>4</sub>. The largest amide shift perturbations between the B-state of S<sub>b4</sub> and B<sub>4</sub> occur for residues proximal to the C-terminus of the B-fold, such as G41, where S<sub>b4</sub> has additional residues and B<sub>4</sub> does not. Many of the S<sub>b4</sub> signals also matched well with S<sub>b3</sub>, although the degree of similarity was not as extensive as with B<sub>4</sub> (Supplementary Fig. 12D–F). More significant amide chemical shift differences between the S-state of S<sub>b4</sub> and S<sub>b3</sub> are likely due to the Y5L mutation, which is a relatively large change located adjacent to the core. To resolve these ambiguities, backbone resonance assignments were made for the S-state of S<sub>b4</sub> (Fig. 7a, [https://doi.org/10.13018/BMR51719] see the "Methods" section for details). Comparison of S<sub>b4</sub> S-state assignments with S<sub>b3</sub> indicated that most of the larger amide shift perturbations were in the β1 and β4 strands. Secondary shift analysis showed that the pattern of secondary structure elements for the S-state of S<sub>b4</sub> is similar to that of S<sub>b3</sub> (Fig. 7b). Inter-proton NOE analysis indicated that the arrangement of the β-strands is also similar (Fig. 7c). Together, these results show that S<sub>b4</sub> populates both S- and B-folds approximately equally at 25 °C. Moreover, a ZZ-exchange spectrum demonstrated that the S- and B-states of S<sub>b4</sub> are in slow conformational exchange on the NMR timescale (Fig. 7d).

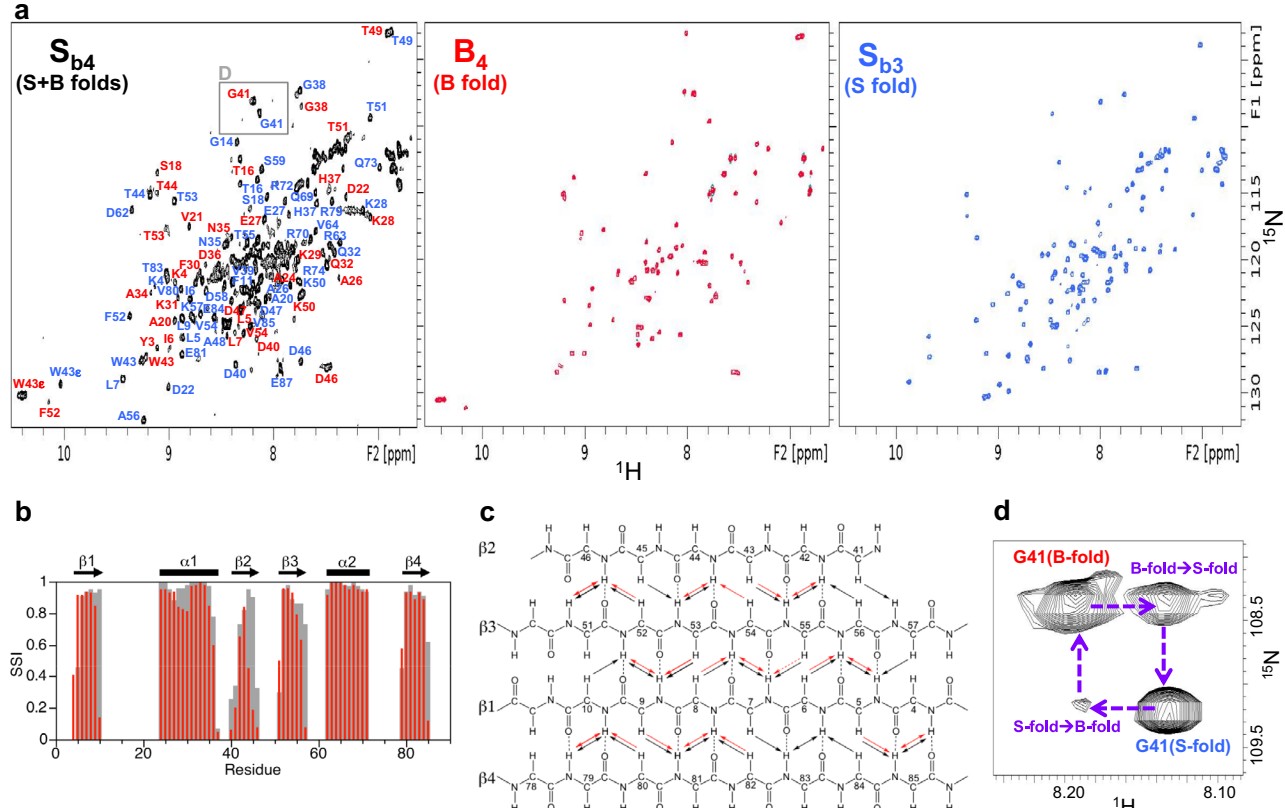

**Fig. 7 | S$_{b4}$ is an equilibrium mixture of S- and B-states. a** Two-dimensional $^1$H–$^{15}$N HSQC spectra of S$_{b4}$ (left), B$_4$ (center), and S$_{b3}$ (right) at 25 °C. In the S$_{b4}$ spectrum, backbone amide resonance assignments are shown for the S-state (blue) and the B-state (red). **b** Confidence levels of ordered secondary structure regions for the S-state of S$_{b4}$ at 30 °C (red) and for S$_{b3}$ at 25 °C (gray) from chemical shifts using TALOS-N. The secondary structure elements of S$_{b3}$, determined from the three-dimensional structure, are shown above the plot. **c** Summary of long-range backbone NOEs from 3D $^{15}$N-NOESY data for S$_{b3}$ at 25 °C (black) and the S-state of S$_{b4}$ at 30 °C (red). **d** Gly41 region of the 300 ms ZZ-exchange spectrum for S$_{b4}$ at 25 °C, showing exchange peaks between the S- and B-states.

Finally, we designed a mutation of leucine 67 to arginine (L67R) in S$_{b4}$ to destabilize the S-fold without changing the sequence of the embedded B-fold. The mutant is denoted as S$_{b5}$ (Supplementary Fig. 10). This was expected to shift the population to the B-fold. The 2D $^1$H-$^{15}$N HSQC spectrum of S$_{b5}$ indicates that the L67R mutation does indeed destabilize the S-fold, with the loss of S-type amide cross-peaks and the concurrent appearance of a new set of signals indicating a switch to a B-fold. The superposition of the spectrum of S$_{b5}$ with that of B$_4$ shows that the new signals in S$_{b5}$ largely correspond with the spectrum of B$_4$ (Supplementary Fig. 13). Thus, the L67R mutation shifts the equilibrium from the S-fold to the B-fold. The additional signals (~25–30) in the central region of the HSQC spectrum that are not detected in B$_4$ are presumably due to the disordered C-terminal tail of S$_{b5}$. The C-terminal tail of S$_{b5}$ does not appear to interact extensively with the B-fold, as evidenced by few changes in chemical shifts or peak intensities in the B-region of S$_{b5}$ compared with B$_4$.

### Structural comparison of S$_{b3}$ and B$_4$

The aligned amino acids 1–56 of S$_{b3}$ and B$_4$ have 98% sequence identity, the only difference being an L5Y mutation in S$_{b3}$ (Fig. 6a). The global folds of S$_{b3}$ and B$_4$ have large-scale differences, however (Fig. 8a, Supplementary Fig. 4). The β1-strands, while similar in length, are in opposite directions in S$_{b3}$ and B$_4$. The β1-strand forms a parallel-stranded interaction with β4 in B$_4$, but an antiparallel interaction with the corresponding β3-strand in S$_{b3}$. Whereas residues 9-20 form the 6-residue β1–β2 turn and the 6-residue β2-strand of B$_4$, these same amino acids constitute the end of β1 and 10 residues of the largely disordered β1–α1 loop in S$_{b3}$. The remainder of the B-region is topologically similar, with the α1/β3/β4 structure in B$_4$ matching the α1/β2/β3 structure in S$_{b3}$.

Overall, however, the order of H-bonding in the 4-stranded β-sheets is quite different, with β2β3β1β4 in S$_{b3}$ and β3β4β1β2 in B$_4$.

The main core residues of B$_4$ consist of Y3, L5, L7, and L9 from β1, A26, F30, and A34 from α1, and F52 and V54 from β4 (Fig. 8b). In S$_{b3}$, the topologically equivalent regions of the core are A26, F30, and A34 from α1, and F52 and V54 from β3. Residues Y5, L7, and L9 from the β1 strand of S$_{b3}$ also form part of the core, but with different packing from B$_4$ due to the reverse orientation of β1. Residues A12 and A20, which contribute to the periphery of the core in B$_4$, are solvent accessible in the β1-α1 loop of S$_{b3}$. Most of the remaining core residues of S$_{b3}$ come from outside of the B-region and include amino acids from β3 (A56), α2 (V64, L67, A68, L71), and β4 (V80 and I82).

### Energetics of unfolding for B$_3$/S$_{b3}$, B$_4$/S$_{b4}$, and S$_{b5}$

Far-UV CD spectra were measured for B$_3$, B$_4$, S$_{b3}$, S$_{b4}$, and S$_{b5}$ and their thermal unfolding profiles were determined by measuring ellipticity at 222 nm versus temperature (Fig. 5, Supplementary Fig. 10, Supplementary Table 1). As described above, the predominant form of S$_{b3}$ is an S-fold. CD and NMR analyses show that B$_3$ is predominantly a B-fold with a Δ$G_{folding}$ of −1.2 kcal/mol at 25 °C[58]. From the NMR analysis, it appears that the B-fold is in equilibrium with putatively dimeric states. This creates a situation in which the B-fold is both temperature-dependent and concentration-dependent. The predominant form at 5 °C and ≤18 μM is the B-fold, however. The low stability and concentration-dependent behavior of B$_3$ may indicate that some propensity for S-type conformations could persist in the 56-residue protein.

S$_{b4}$ has a temperature unfolding profile very similar to S$_{b3}$ (Fig. 5) even though both S- and B- are approximately equally populated at 25 °C in S$_{b4}$ (Fig. 7). This shows that the Y5L mutation results in two folds that

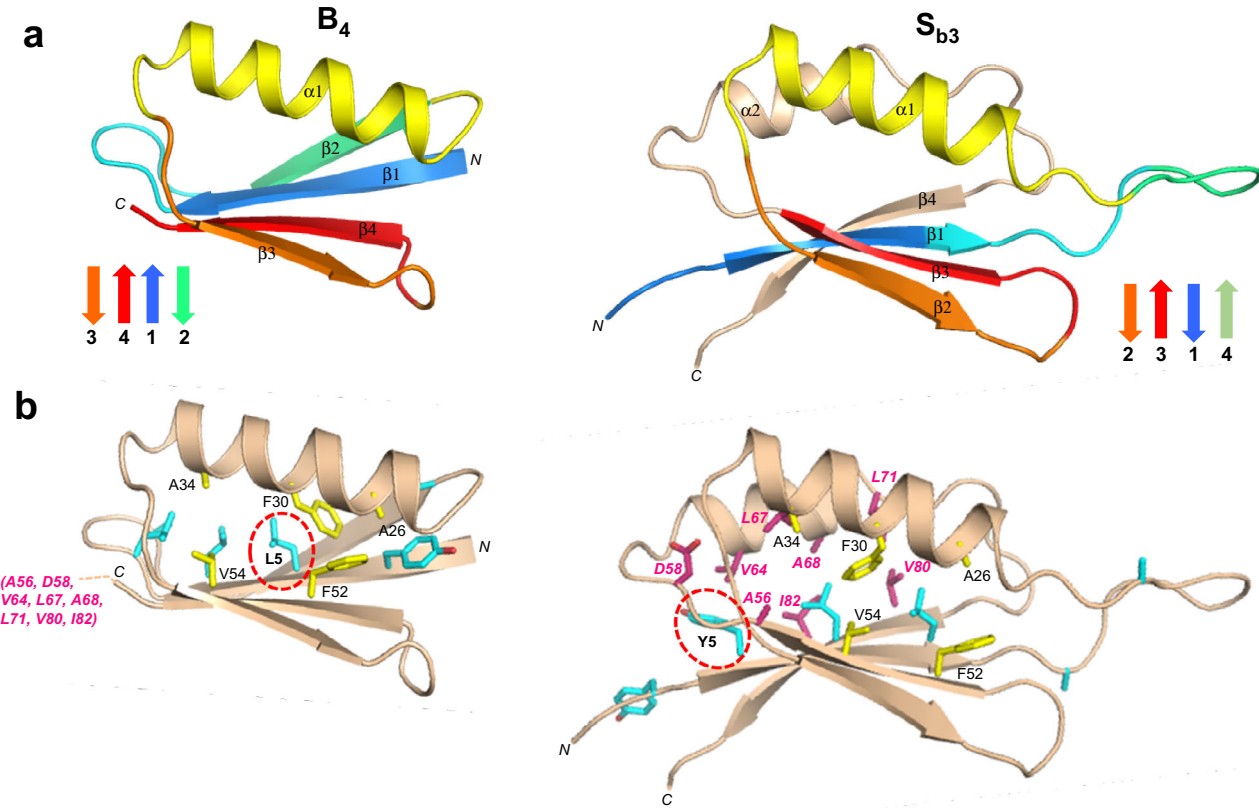

**Fig. 8 | Structural differences in the high (~98%) sequence identity regions of B₄ and S_{b3}. a** Main chain comparisons. (Left panel) CS-Rosetta structure of B₄ with secondary structure elements color coded. (Right panel) Corresponding color-coded regions mapped onto the CS-Rosetta structure of S_{b3}, showing changes in backbone conformation. Regions outside the 56 amino acid sequence of B₄ are shown in wheat. **b** Side chain comparisons. (Left panel) Residues contributing to the core of B₄ from α1/β3/β4 (yellow), and from other regions (cyan). The non-α1/β2/β3 core residues from S_{b3} (pink) do not overlap with the B₄ core (see text for further details). (Right panel) Residues contributing to the core of S_{b3} from α1/β2/β3 (yellow), and the other participating core residues (pink). The non-α1/β2/β3 core residues from B₄ are also shown (cyan). The single L5Y amino acid difference between B₄ and S_{b3} is highlighted.

are almost isoenergetic and both thermodynamically stable relative to the unfolded state. Further, because S- and B-folds are in equilibrium and approximately equally populated, the free energy of switching to the B-fold from the S-fold ($\Delta G_{\text{B-fold/S-fold}}$) is ~0 kcal/mol at 25 °C. The switch equilibrium reflects the influence of the antagonistic B-fold on the S-fold population in S_{b4}, where the leucine at residue 5 helps stabilize the alternative B-state at the expense of the S-state. Thermal denaturation by CD shows that B₄ has a $\Delta G_{\text{folding}} = -4.1$ kcal/mol at 25 °C[58]. The thermal unfolding profile of S_{b5} shows a low-temperature transition with a midpoint ~10 °C and a major transition with a midpoint of ~60 °C (Fig. 5b). The NMR analysis indicates that the major transition is unfolding of the B-fold. Thus, the arginine at 67 in S_{b5} makes the B-fold more favorable by making the S-fold unfavorable, consistent with the change in population from mixed to B-fold observed by NMR.

### Protease inhibition

The S_{b3} protein is closely related to S′I but lacks inhibitor function because C-terminal amino acids were changed in the design of the switch. It can be converted into a protease inhibitor, however, by altering C-terminal amino acids VTE to DKLYRAL. This mutant is denoted S_{b3}I. S_{b3} and S_{b3}I appear similar in structure by CD analysis (Supplementary Fig. 10). The K_I for S_{b3}I with the engineered subtilisin was determined to be 50 nM (Supplementary Table 1).

### IgG binding

Binding to IgG was determined for B₃ and S_{b3}I (Supplementary Table 1). B₃ and S_{b3}I bound to IgG Sepharose with $K_D \leq 1\,\mu M$ and $10\,\mu M$, respectively. Presumably, S_{b3}I has significant IgG-binding

activity because the α1β3 IgG binding surface of the B-fold is largely preserved in the S-fold. Thus, S_{b3}I is a dual-function protein with both IgG-binding and protease inhibitor functions (Fig. 2f).

## Discussion

The entire network of intersecting pathways between the S-, A-, and B-folds is summarized in Fig. 9. The first node on the pathway is a functional switch from RNA binding protein to protease inhibitor without a fold switch. The α/β plait is a common fold, and proteins with this basic topology include many different functions[42]. Engineering the SI and S′I nodes illustrates how protease inhibitor function can arise in the α/β plait topology with a few mutations. Replacing only C-terminal amino acids in the S6 protein creates interaction with the substrate binding cleft of the protease (Fig. 2a, b). This C-terminal interaction plus adventitious contact between the β-sheet surface of the α/β plait and two α-helices in the protease result in protease inhibition in the 50 nM range. Based on the structure of S6 in the 30S complex, the C-terminal modification may not have major effects on binding interactions with ribosomal RNA and the S15 protein (Fig. 2a)[43]. Thus, the transition from RNA binding protein to protease inhibitor likely is uninterrupted. An insertion in the β1–α1 loop and a deletion β2–β3 loop in the SI-inhibitor creates a topology that more closely resembles natural prodomain-type inhibitors[44,46,61] and creates an α1β2β3 motif in the S′-fold that is similar to the α1β3β4 motif of the B-fold. This topological similarity brings the S′I closer to an intersection with the B-fold. Thus, SI and S′I nodes are both functional switches and branch points for switching the S-fold into the A- and B-folds, respectively.

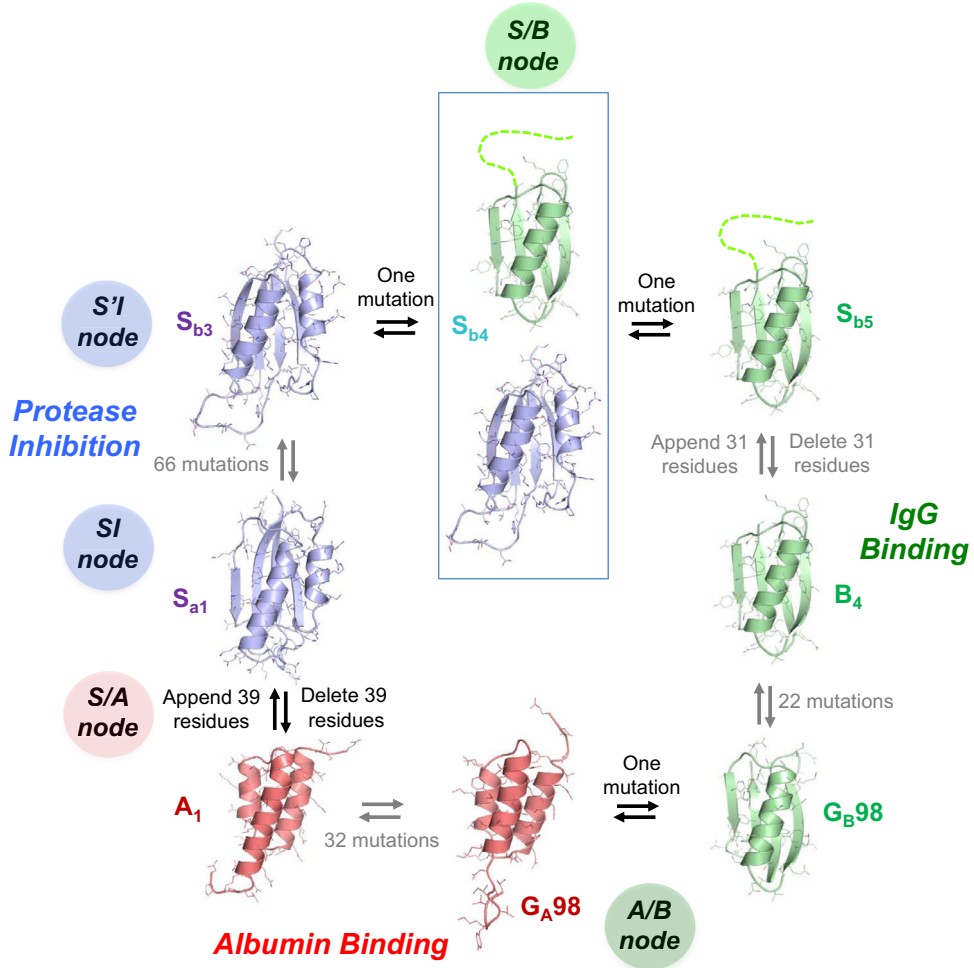

**Fig. 9 | Sequence-fold relationships of engineered S/A, S/B, and A/B nodes.** Switches between stable folds can be induced by a single amino acid mutation or deleting/appending terminal sequence that stabilizes the S-fold. Blue denotes an S-fold, green a B-fold, and red an A-fold. Gray arrows connect proteins that have been reengineered without a fold switch. $S_{b4}$ is observed with two folds simultaneously. The $G_A98$ and $G_B98$ structures are from PDB codes 2LHC and 2LHD (ref. [32]), respectively.

Engineering nodes at fold intersections required designing sequences that are compatible with native interactions in two different folds. We used simple rules to do this. The first rule was to align topologies rather than maximizing sequence similarities. Identifying a common topology can help determine a register that has fewer irreconcilable clashes. For example, topological alignment of the α1 helix of the SI fold and the α1 helix of the A-fold facilitated engineering the fold switch, because the regions flanking α1 of the SI-fold can encode two different fold motifs. When topological alignment is poor, as was the case with S- and B-folds, it was helpful to look for natural variations in the turns of the longer fold to create better alignment. Variation in loops and turns in a larger fold creates more freedom of design and a higher probability of switches. Once an alignment is chosen, the basic rule in resolving catastrophic clashes is to conserve original amino acids when possible. This reduces the uncertainties involved in computational design. The Rosetta energy function was not used to predict a favorable alignment but was important in evaluating mutations to resolve clashes once an alignment was chosen.

Selecting mutations compatible with two sets of native interactions required tradeoffs in the native state energetics of each individual fold[5,11]. A node may be produced in cases in which both alternative folds are stable relative to the unfolded state. Stability relative to the unfolded state (i.e. a state with little secondary structure) was determined by CD melting (Fig. 5). It was informative to examine the stability of both short (56 residues) and longer forms of a putative node sequence. The independent stability of the G-fold can be determined in the short form without the antagonism from the S-fold that is present in the longer sequence. The stabilities of the $A_1$ and $A_2$ proteins are about −4 kcal/mol at 25 °C[58] compared to −5.6 kcal/mol for the native $G_A$ protein[56]. The stabilities of $B_3$ and $B_4$ are −1.2 and −4.1 kcal/mol, respectively, at 25 °C[58] compared to −6.7 kcal/mol for the native $G_B$ protein[62]. For the longer sequences, the $\Delta G_{folding}$ of $S_{a1}$ and $S_{b3}$ are −5.3 and −3.5 kcal/mol, respectively, at 25 °C[58] compared to −8.5 kcal/mol for the native S6 protein[40].

In the case of the S-folds, however, the energetic effects of the stable, embedded G-fold must also be considered. Since the equilibria between both folded states and the unfolded state are thermodynamically linked, the free energy of a switch to a G-fold from an S-fold ($\Delta G_{\text{G-fold/S-fold}}$) is approximated by the difference in $\Delta G_{folding}$ ($\Delta\Delta G_{folding}$) between the short and long forms of a node protein. For example, based on $\Delta G_{folding}$ for $A_1$ and $S_{a1}$, the predicted $\Delta G_{\text{A-fold/S-fold}}$ of $S_{a1}$ is 1.3 kcal/mol. This is consistent with the structure of the predominant S-fold determined by NMR but also with the small population of 3α fold suggested by weak HSA-binding. From the thermal denaturation profiles of $B_3$ and $S_{b3}$, the predicted $\Delta G_{\text{B-fold/S-fold}}$ of $S_{b3}$ is 2.3 kcal/mol, a value consistent with the stable S-fold observed in NMR experiments. The $S_{b3}$ sequence is also approaching a critical point, however. A substitution in $S_{b3}$ that stabilizes the B-fold (Y5L) shifts the equilibrium of $S_{b4}$ to an approximately equal mixture of B- and S-folds.

That is, $\Delta G_{\text{B-fold/S-fold}}$ of S$_{b4}$ is -0 kcal/mol at 25 °C. One further substitution that destabilizes the S-fold (L67R) shifts the population of S$_{b5}$ to a stable B-fold ($\Delta G_{\text{B-fold/S-fold}} \leq -5$ kcal/mol) (Fig. 9).

The existence of nodes between folds has implications for the evolution of new functions. In the case of the S/A node, all contact amino acids for HSA exist within the S-fold of the protease inhibitor S$_{a2}$I albeit in a cryptic topology. Deletion of amino acids 67–99 (A$_2$) results in loss of inhibitor function and a fold switch from α/β plait to 3α. Acquisition of HSA binding activity ($K_D < 1\,\mu M$) results from unmasking the cryptic HSA binding amino acids via the fold switch (Fig. 2e). This level of binding affinity could be biologically relevant since the concentration of HSA in serum is $>500\,\mu M$[63]. In the case of the S/B node, the α1β3 motif contains all IgG contact amino acids and S$_{b3}$I has some affinity for both IgG ($K_D = 10\,\mu M$) and protease ($K_I = 50$ nM). In this case, the Y5L mutation (S$_{b4}$) or a deletion of 57–91 (B$_4$) causes a fold switch from α/β plait to the β-grasp and results in tighter IgG binding ($K_D \leq 1\,\mu M$) (Fig. 2f). This level of binding affinity could also be biologically relevant since the concentration of IgG in serum is $>50\,\mu M$ (or $>100\,\mu M$ Fc binding sites)[64]. We have previously shown that an A-fold with HSA binding function can be switched to a B-fold with IgG-binding function via single amino acid substitutions that switch the folds and unmask cryptic contact amino acids for the two ligands[29,32].

In conclusion, it was possible to connect three common folds in a network of high-identity nodes that form critical points between two folds. As in other complex systems, a small change in a protein near a critical point can have a "butterfly effect" on how the folds are populated. This property of the protein folding code means that proteins with multiple folds and functions can exist in highly identical amino acid sequences. This suggests that the evolution of new folds and functions sometimes can follow uninterrupted mutational pathways.

## Methods

### Mutagenesis, protein expression and purification
Mutagenesis was carried out using Q5® Site-Directed Mutagenesis Kits (NEB). G$_A$ and G$_B$ variants were cloned into a vector (pH0720) encoding the sequence:

MEAVDANSLA QAKEAAIKEL KQYGIGDKYI KLINNAKTVE GVESLKNEIL KALPTEGSGN TIRVIVSVDK AKFNPHEVLG IGGHIVYQFK LIPAVVVDVP ANAVGKLKKM PGVEKVEFDH QYRGL

as an N-terminal fusion domain[56]. Cell growth was carried out by auto-induction[29,65]. Cells were harvested by centrifugation at $3750 \times g$ for 20 min and lysed by sonication on ice in 0.1 M KPi, pH 7.2. Cellular debris was pelleted by centrifugation at $10,000 \times g$ for 15 min. Supernatant was clarified by centrifugation at $45,000 \times g$ for 30 min. Proteins were purified using a second generation of the affinity-cleavage tag system employed previously to purify switch proteins[29,66]. The second-generation tag results in high-level soluble expression of the switch proteins and also enables the capture of the fusion protein by binding tightly to an immobilized processing protease via the C-terminal EFDHQYRGL sequence. Loading and washing were at 5 mL/min for a 5 mL *Im-Prot* column using a running buffer of 20 mM KPi, pH 6.8. The amount of washing required for high purity depends on the stickiness of the target protein and how much of it is bound to the column. We typically wash with 10 column volumes (CV) of wash solution followed by 3 CV 0.5 M NaCl and then -10 CV running buffer. This can be repeated as necessary. The 0.5 M NaCl shots are repeated until the amount of absorbance released with each high-salt shot becomes small and constant. All the high-salt solution is washed out before initiating the cleavage. The target protein was cleaved from the *Im-Prot* column by injecting 15 mL of imidazole solution (0.1 mM) at 1 mL/min, 22 °C. The cleaved protein typically elutes as a sharp peak in 2–3 CV. The purified protein was then concentrated to 0.2–0.3 mM, as

required for NMR analysis. The columns were regenerated by injecting 15 mL of 0.1 N H$_3$PO$_4$ (0.227 mL concentrated phosphoric acid (85%) per 100 mL) at a flow rate of -1 CV/min. The wash solution was neutralized immediately after stripping. The purification system is available from Potomac Affinity Proteins.

Protease inhibitor proteins were purified by binding to *Im-Prot* media and then stripping off the purified inhibitor with 0.1 N H$_3$PO$_4$. Samples were then immediately neutralized by adding 1/10 volume 1 M K$_2$HPO$_4$.

### Rosetta calculations
Rosetta energies of all designed structures were generated using the Slow Relax routine[54]. 1000 decoys were calculated for each design. PDB coordinates and energy parameters for the lowest energy decoy for each design are included as supplemental files.

### Circular dichroism (CD)
CD measurements were performed in 100 mM KPi, pH 7.2 with a Jasco spectropolarimeter, model J-1100 with a Peltier temperature controller. Quartz cells with path lengths of 0.1 and 1 cm were used for protein concentrations of 3 and 30 μM, respectively. The ellipticity results were expressed as mean residue ellipticity, $[\theta]$, deg cm$^2$ dmol$^{-1}$. Ellipticities at 222 nm were continuously monitored at a scanning rate of 0.5°/min. Reversibility of denaturation was confirmed by comparing the CD spectra at 20 °C before melting and after heating to 100 °C and cooling to 20 °C.

### Measuring HSA and IgG binding affinity
Affinity of proteins to HSA and IgG was determined by their retention on the immobilized ligands. HSA and rabbit IgG were immobilized by reaction with NHS-activated Sepharose 4 Fast Flow (Cytiva) according to the manufacturer's instructions. The concentration of immobilized HSA was 100 μM. The concentration of immobilized IgG was 50 μM (i.e. 100 μM Fc binding sites). Generally, 0.2 mL of a 5 μM solution of the test protein was injected into a 5 mL column at a flow rate of 0.5 mL/ min. Determination of binding affinity assumes that binding is in rapid equilibrium such that the elution volume is proportional to the fraction of test protein bound to 100 μM of binding sites. Proteins that are completely retained after 20 column volumes (CV) are assessed to have $K_D \leq 1\,\mu M$. Completely retained proteins are stripped from the column with 0.1 N H$_3$PO$_4$ at the end of the run.

### Measuring protease inhibition
Competitive inhibition constants ($K_I$) were determined using the fluorogenic peptide substrate QEEYSAM-AMC (7-amino-4-methylcoumarin) purchased from AnaSpec Inc. and a highly specific, engineered protease known as RASProtease(I)[49]. Competitive inhibition constants ($K_I$) were measured by determining the $K_{\text{M(apparent)}}$ in the presence of 0, 50, and 100 nM of each inhibitor protein. The reactions were carried out in 100 mM KPi, 10 mM imidazole, 0.005% tween-20, pH 7.0 at 25 °C with 1 nM RASProtease(I). The QEEYSAM-AMC concentrations used to determine $K_M$ and $K_{\text{M(apparent)}}$ were 0.1, 0.5, 1, 2, 5, and 10 μM. Initial rates were determined with a BioTek Synergy MT fluorescence microplate reader (Ex: 360/40, Em: 460/40) by measuring the release of the fluorescent AMC group via hydrolysis of the amide bond. Highly pure (≥98%) protease and inhibitor proteins were used for all kinetic experiments.

### NMR spectroscopy
Isotope-labeled samples were prepared at 0.2–0.3 mM concentrations in 100 mM potassium phosphate buffer (pH 7.0) containing 5% D$_2$O. NMR spectra were collected using Topspin3.6.1 software on Bruker AVANCE III 600 and 900 MHz spectrometers fitted with Z-gradient $^1$H/ $^{13}$C/$^{15}$N triple resonance cryoprobes. Standard double and triple resonance experiments (HNCACB, CBCA(CO)NH, HNCO, HN(CA)CO, and

HNHA) were utilized to determine main chain NMR assignments. Inter-proton distances were obtained from 3D $^{15}$N-edited NOESY and 3D $^{13}$C-edited NOESY spectra with a mixing time of 150 ms. NmrPipe[67] was used for data processing and analysis was done with Sparky[68]. Two-dimensional {$^{1}$H}-$^{15}$N steady-state heteronuclear NOE experiments were acquired with a 5 s relaxation delay between experiments. Errors in heteronuclear NOEs were estimated based on the background noise level. Chemical shift perturbations were calculated using $\Delta\delta_{total} = ((W_H\Delta\delta_H)^2 + (W_N\Delta\delta_N)^2)^{1/2}$, where $W_H$ is 1, $W_N$ is 0.2, and $\Delta\delta_H$ and $\Delta\delta_N$ represent $^{1}$H and $^{15}$N chemical shift changes, respectively. For PRE experiments on $S_{b1}$, single-site cysteine mutant samples were incubated with 10 equivalents of (1-oxyl-2,2,5,5-tetramethylpyrroline-3-methyl) methanethiosulfonate (MTSL, Santa Cruz Biotechnology) at 25 °C for 1 h and completion of labeling was confirmed by MALDI mass spectrometry. Control samples were reduced with 10 equivalents of sodium ascorbate. Backbone amide peak intensities of the oxidized and reduced states were analyzed using Sparky. Three-dimensional structures were calculated with CS-Rosetta3.2 using experimental backbone $^{15}$N, $^{1}$H$_N$, $^{1}$H$\alpha$ $^{13}$C$\alpha$, $^{13}$C$\beta$, and $^{13}$CO chemical shift restraints and were either validated by comparison with experimental backbone NOE patterns ($A_1$, $B_1$, $B_4$, $S_{b1}$) or directly employed interproton NOEs ($S_{a1}$, $S_{b2}$) or PREs ($S_{b1}$) as additional restraints. One thousand CS-Rosetta structures were calculated from which the 10 lowest energy structures were chosen. For $S_{b3}$, CS-Rosetta failed to converge to a unique low-energy topology, producing an approximately even mixture of S- and B-type folds despite the chemical shifts and NOE pattern indicating an S-fold. In this case, CNS1.1[69] was employed to determine the structure[56], including backbone dihedral restraints from chemical shift data using TALOS-N[70]. The backbone resonances for the S-state of $S_{b4}$ were assigned using triple resonance methods as above, under conditions where the S-state is more favorably populated (30 °C, 100 mM KPi, 200 mM sodium chloride, pH 7.0). Amide assignments were then transferred to the two-dimensional $^{1}$H-$^{15}$N HSQC spectrum of $S_{b4}$ at 25 °C in 100 mM KPi, pH 7.0. Inter-proton NOEs for the S-state of $S_{b4}$ were obtained at the 30 °C/high salt condition, employing a 3D $^{15}$N-edited NOESY spectrum with a 150 ms mixing time. A two-dimensional ZZ-exchange $^{1}$H–$^{15}$N HSQC spectrum was recorded on $S_{b4}$ using a mixing time of 300 ms (25 °C, 100 mM KPi, pH 7.0)[71,72]. Protein structures were displayed and analyzed utilizing PROCHECK-NMR[73], MOLMOL[74] and PyMol (Schrodinger)[55].

### Reporting summary

Further information on research design is available in the Nature Portfolio Reporting Summary linked to this article.

## Data availability

The NMR structures generated in this study have been deposited in the PDB: [https://doi.org/10.2210/pdb7MN1/pdb]; [https://doi.org/10.2210/pdb7MQ4/pdb]; [https://doi.org/10.2210/pdb7MN2/pdb]; [https://doi.org/10.2210/pdb7MP7/pdb]; [https://pdb-dev.wwpdb.org/entry.html?PDBDEV_00000083]; [https://pdb-dev.wwpdb.org/entry.html?PDBDEV_00000084]; [https://pdb-dev.wwpdb.org/entry.html?PDBDEV_00000085]. NMR Assignments have been deposited in the BMRB: [https://doi.org/10.13018/BMR30901]; [https://doi.org/10.13018/BMR30902]; [https://doi.org/10.13018/BMR30904]; [https://doi.org/10.13018/BMR30905]; [https://doi.org/10.13018/BMR50907]; [https://doi.org/10.13018/BMR50909]; [https://doi.org/10.13018/BMR50910]; [https://doi.org/10.13018/BMR51719]. The structures referenced in this paper are publicly available in the PDB: [https://doi.org/10.2210/pdb1FKA/pdb]; [https://doi.org/10.2210/pdb2VDB/pdb]; [https://doi.org/10.2210/pdb1FCC/pdb]; [https://doi.org/10.2210/pdb6UAO/pdb]; [https://doi.org/10.2210/pdb2LHC/pdb]; [https://doi.org/10.2210/pdb1RIS/pdb]. Source data are provided with this paper. Design models are provided as files in the source data. Source data are provided with this paper.

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

## Acknowledgements

This work was supported by National Institutes of Health Grant GM62154 (to P.B. and J.O.) and 5R44GM126676 (to P.B.). The NMR facility is supported by the University of Maryland, the National Institute of Standards and Technology, and a grant from the W. M. Keck Foundation. We also thank Drs. Nese Sari and Louisa Wu for critically reading the manuscript and for many thoughtful comments. Mention of commercial products does not imply recommendation or endorsement by NIST.

## Author contributions

Protein design: Yw.C., B.R., E.C., J.O., P.B.; Performed thermodynamic and binding analyses: B.R., Yw.C., D.M., R.S., P.B.; Performed dynamic light scattering experiments: T.G.; Performed NMR experiments/ structural analysis: Y.H., Yh.C., T.S., T.K., J.O.; Wrote the paper: J.O. (NMR and structural analysis), Yw.C., B.R., P.B. (remaining sections).

## Competing interests

The authors declare no competing interests.
