## [Peer Review File · Nature Communications]

REVIEWER COMMENTS

Reviewer #1 (Remarks to the Author):

I think this is a very impressive manuscript that deals with a fascinating and important problem of protein fold changes in evolution of protein structure and function. These processes are not very well understood, and this work presents illustrious examples of transitions among 3 folds and their functions by means of point mutations and insertions/deletions at the protein termini. These transitions are achieved by targeted design rather than artificial evolution, but the steps involved are within the framework of standard evolutionary events, so it is conceivable that such transitions could happen in evolution. The design examples presented are rather remarkable, e.g., that long sequence that changes its conformation (except the central helix) after addition of peptides at its termini. Although such transitions were known for shorter sequences ("chameleon" sequences), the examples presented in this paper push the envelope and are certainly textbook-worthy.

Minor comments:

- 1. Although the paper is understandable to specialists, just because the examples are so remarkably interesting it would be nice if the authors can possibly "popularize" their work by making it a little more accessible to broader audience. Here is a simple example. Their Fig. 2 is very good, and yet I'm sure that some readers may stumble at "Y5L", etc. The authors state "66 mutations", "22 mutations", then why not say "1 mutation"? It is not obvious for outsiders that "Y5L" means 1 mutation. The authors state "Append 39". Why not say "Append 39 residues". Again, such small changes will improve the accessibility of these wonderful illustrations.**
- 2. I wouldn't call beta-grasp a "most common" fold. It is actually much less common than the other two.**
- 3. "The structures of proteins around nodes were determined using NMR spectroscopy". I didn't find PDB accession codes for these structures in the manuscript. Experimental structures should be deposited in PDB, codes given in the main text of the paper, and PDBs released immediately upon publication.**
- 4. Previous fold switching experiments relied on point mutations. Here, a significant difference is that N- and C- terminal extensions or truncations may be the most important components of the switching network. This is not clearly articulated or stressed (although mentioned, and rather clearly), but I think it should be, even in the abstract. Otherwise it sounds a bit like "cheat": not what a naïve reader may think by reading the abstract. There is nothing wrong with using extensions because indels are routine.**

Reviewer #2 (Remarks to the Author):

Ruan et al describe the design of a series of closely-related sequences (a 'protein fold switching network') that adopt one of three different folds, A (helical bundle), B and S (both mixed helix/sheet folds). This is the latest installment in a longstanding project to understand the origins and design of metamorphic and fold-switching proteins. Whereas previous studies used phage display to merge the A and B folds into a single sequence with

the desired properties, here they perform *in silico* optimization using Rosetta to design sequences that encoded two folds simultaneously. The presence or absence of N- and/or C-terminal flanking sequences is used to favor one or the other of two distinct structures, and function-specific substitutions are used to restore binding interfaces for albumin or IgG binding or protease inhibition. The experimental data are of high quality and the description of approaches and interpretation of the results are adequate in most instances. This paper represents an important advance in the design of fold switching proteins but has some flaws that must be remedied.

In particular, what appears to be the most significant result is largely ignored. Sb4 appears to be a protein that simultaneously occupies both the S and B folds in equal proportions. This is quite interesting since it may be the first truly metamorphic designed protein and deserving of more scrutiny. The HSQC spectrum should be included in Fig S10 and its unfolding curve added to Fig 6. Additionally, an ¹⁵N-zz exchange experiments should be performed to measure the kinetics of interconversion.

The authors are mistaken in stating that "the stability of Sb4 cannot be derived from the NMR analysis". The relative proportions report only on the $\Delta\Delta G$, or difference in thermostabilities of the two folds in the chosen environment, not the overall ΔG of folding. The unfolding should be measured by CD as it exhibits a strong helical signature comparable to the other Sb(n) variants.

A minor concern: it is stated several times that the convergent designs populate the three most common small folds with no supporting evidence. The taxonomy and frequency of protein folds has been catalogued in detail for at least a few decades, and a suitable review article with a recent histogram of genome-wide fold frequency should be cited.

The discussion recaps the iterative design process but lacks a statement of purpose (the abstract subheading 'Engineering intersecting pathways between folds and functions' is insufficient) leaving any broad impact of the study unstated. More importantly, the authors seem to miss a prominent feature of their results when tabulating rules for design of fold-switching proteins: reduced thermostability. Selection for relatively unstable folding has been noted previously as a key criterion in the design of fold-switching proteins (e.g. Dishman 2022 *Curr Opin Struct Biol*), and the values in Table S1 reinforce this idea. The parent sequences SI, Sa1I and S'I fold with deep wells (~ 8 kcal/mol) in the folding energy landscape and are steadily evolved into sequences with considerably lower ($\sim 1-4$ kcal/mol) thermostabilities.

Ruan et al. response to reviewers' comments

We appreciate the thorough review and many insightful comments. We have addressed all points as detailed below.

REVIEWER COMMENTS

Reviewer #1 (Remarks to the Author):

Minor comments:

1. Although the paper is understandable to specialists, just because the examples are so remarkably interesting it would be nice if the authors can possibly "popularize" their work by making it a little more accessible to broader audience.

Here is a simple example. Their Fig. 2 is very good, and yet I'm sure that some readers may stumble at "Y5L", etc. The authors state "66 mutations", "22 mutations", then why not say "1 mutation"? It is not obvious for outsiders that "Y5L" means 1 mutation. The authors state "Append 39". Why not say "Append 39 residues". Again, such small changes will improve the accessibility of these wonderful illustrations.

Response: *We have reorganized the introduction and rewritten the discussion. We have simplified language throughout. We have modified Figure 2 as suggested (It is now Figure 8).*

2. I wouldn't call beta-grasp a "most common" fold. It is actually much less common than the other two.

Response: *We have removed the phrase "three most common folds". We have included a reference to most populated folds and added lines 63-65:*

These proteins share no significant homology and are representative of three of the ten most common folds: the S-fold is a thioredoxin-like α/β plait; the A-fold is a homeodomain-like 3α -helix bundle; the B-fold is a ubiquitin-like β grasp³².

3. "The structures of proteins around nodes were determined using NMR spectroscopy". I didn't find PDB accession codes for these structures in the manuscript. Experimental structures should be deposited in PDB, codes given in the main text of the paper, and PDBs released immediately upon publication.

Response: *PDB codes are include in Tables 1 and 2 and also now noted in the text. The structures are available in the Protein Data Bank.*

4. Previous fold switching experiments relied on point mutations. Here, a significant difference is that N- and C- terminal extensions or truncations may be the most important components of the switching network. This is not clearly articulated or stressed (although mentioned, and rather clearly), but I think it should be, even in the abstract. Otherwise it sounds a bit like "cheat": not what a naïve reader may think by reading the abstract. There is nothing wrong with using extensions because indels are routine.

Response: We agree and have rewritten the abstract to make this clear.

Reviewer #2 (Remarks to the Author):

1. In particular, what appears to be the most significant result is largely ignored. Sb4 appears to be a protein that simultaneously occupies both the S and B folds in equal proportions. This is quite interesting since it may be the first truly metamorphic designed protein and deserving of more scrutiny.

1A. The HSQC spectrum should be included in Fig S10 and its unfolding curve added to Fig 6.

Response: The unfolding curve for Sb4 was added to Fig 5. The HSQC spectrum is included in Fig S12 and compared to spectra for B4 and Sb3.

1B. Additionally, an ^{15}N -zz exchange experiments should be performed to measure the kinetics of interconversion.

Response: A ^{15}N -zz exchange experiment has been performed and establish that the two folds are in equilibrium on the s^{-1} time scale. We have also attached a region of the ^{15}N -zz exchange spectrum below for the reviewer's inspection.

Figure: 2D ^1H - ^{15}N HSQC and zz-exchange spectra of Sb4 at 25°C, highlighting the exchange peaks between the B-fold and S-fold backbone NH signals of G41. The zz-exchange spectrum was collected with a 300 ms mixing time.

We completely agree that Sb4 is an important milestone. Kinetic analysis of residue-specific interconversion rates (as well as shifts in the S- to B- equilibrium as a function of solvent and mutation) is a large topic on its own and will be the subject of a subsequent paper.

1c. The authors are mistaken in stating that “the stability of Sb4 cannot be derived from the NMR analysis”. The relative proportions report only on the $\Delta\Delta G$, or difference in thermostabilities of

the two folds in the chosen environment, not the overall delta-G of folding. The unfolding should be measured by CD as it exhibits a strong helical signature comparable to the other Sb(n) variants.

Response: These are useful points. The unfolding curve for Sb4 was added to Fig 5, as noted above. We have better described the significance of a $\Delta G_{\text{folding}}$ and its relation to the free energy of a fold switch.

Specifically, the following has been added to the RESULTS (lines 307-311):

S_{b4} has a temperature unfolding profile very similar to S_{b3} even though both B- and S- are equally populated at 25°C in S_{b4} (**Fig. 5**). This shows that the Y5L mutation results in two folds that are almost isoenergetic and both thermodynamically stable relative to the unfolded state. Further, because S- and B-folds are in equilibrium and equally populated, the free energy of switching to the B-fold from the S-fold ($\Delta G_{\text{B-fold/S-fold}}$) is ~ 0 kcal/mol at 25°C.

We have also added the following to the DISCUSSION (lines 384-396):

In the case of the S-folds, however, the energetic effects of the stable, embedded G-fold must also be considered. Since the equilibria between both folded states and the unfolded state are thermodynamically linked, the free energy of a switch to a G-fold from an S-fold ($\Delta G_{\text{G-fold/S-fold}}$) is approximated by the difference in $\Delta G_{\text{folding}}$ ($\Delta \Delta G_{\text{folding}}$) between the short and long forms of a node protein. For example, based on $\Delta G_{\text{folding}}$ for A₁ and S_{a1}, the predicted $\Delta G_{\text{A-fold/S-fold}}$ of S_{a1} is 1.3kcal/mol. This is consistent with the structure of the predominant S-fold determined by NMR but also with the small population of 3 α fold suggested by weak HSA-binding. From the thermal denaturation profiles of B₃ and S_{b3}, the predicted $\Delta G_{\text{B-fold/S-fold}}$ of S_{b3} is 2.3kcal/mol, a value consistent with the stable S-fold observed in NMR experiments. The S_{b3} sequence is also approaching a critical point, however. A substitution in S_{b3} that stabilizes the B-fold (Y5L) shifts the equilibrium of S_{b4} to an approximately equal mixture of B- and S-folds. That is, $\Delta G_{\text{B-fold/S-fold}}$ of S_{b4} is ~ 0 kcal/mol at 25°C. One further substitution that destabilizes the S-fold (L67R) shifts the population of S_{b5} to a stable B-fold ($\Delta G_{\text{B-fold/S-fold}} \leq -5$ kcal/mol) (**Fig. 8**).

2. A minor concern: it is stated several times that the convergent designs populate the three most common small folds with no supporting evidence. The taxonomy and frequency of protein folds has been catalogued in detail for at least a few decades, and a suitable review article with a recent histogram of genome-wide fold frequency should be cited.

Response: We have removed the phrase “three most common folds”. We have included a reference to the most populated folds and added lines 63-65:

These proteins share no significant homology and are representative of three of the ten most common folds: the S-fold is a thioredoxin-like α/β plait; the A-fold is a homeodomain-like 3 α -helix bundle; the B-fold is a ubiquitin-like β grasp³².

3. The discussion recaps the iterative design process but lacks a statement of purpose (the abstract subheading ‘Engineering intersecting pathways between folds and functions’ is insufficient) leaving any

broad impact of the study unstated. More importantly, the authors seem to miss a prominent feature of their results when tabulating rules for design of fold-switching proteins: reduced thermostability. Selection for relatively unstable folding has been noted previously as a key criterion in the design of fold-switching proteins (e.g. Dishman 2022 Curr Opin Struct Biol), and the values in Table S1 reinforce this idea. The parent sequences SI, Sa1I and S'I fold with deep wells (~8 kcal/mol) in the folding energy landscape and are steadily evolved into sequences with considerably lower (~1–4 kcal/mol) thermostabilities.

Response: We have added the reference to Dishman 2022 and rewritten the discussion to cover the energetic implications of designing a sequence with two sets of native interactions and how stability relative to the unfolded state is related to fold switching. This is covered in lines 372-396. We have also added sections on the broader impacts of these results on the evolution of new folds. (Lines 397-411)

REVIEWER COMMENTS

Reviewer #1 (Remarks to the Author):

The paper revised properly according to the reviewers' suggestions. I am satisfied with the revision and do not have more concerns except the abstract may improve to be more friendly to a broad audience of the journal. For example, the first sentence of "Protein sequences encoding three common small folds (3α , β -grasp, and α/β -plait) were connected in a network with high-identity intersections, termed nodes." I can understand the authors want to refer to their Fig 1 but as the first sentence in the abstract, I would like authors to find a more audience-friendly way to explain their concept such as network and high-identity intersections. Does "high identity intersection" refer to "high sequence identity"? But this is more about personal writing style and my suggestion may be quite personal too. Thus, whether the language should be revised would be better decided by the editors who understand the journal audience and target better.

Reviewer #2 (Remarks to the Author):

This revised manuscript is improved and addresses many of the identified weaknesses in the original, but significant concerns detailed below must be addressed before publication.

It is possible that Sb4 is exchanging between the B and S folds; the ZZ-exchange peaks provided for inspection (but not for publication) indicate that some slow conformational change is occurring. However, it is not clear how the assignment of those exchanging peaks was determined to be Gly 41 or which one corresponds to the S and B fold, since the overlay in Fig S12 shows no nearby peaks in the B4 and Sb3 spectra. From the absence of any description of the Sb4 NMR assignments, it appears that they have not been determined unambiguously. If so, the conclusion that Sb4 exchanges between the B and S folds (e.g. Fig. 8 and lines 395-6) is not adequately supported by the results presented in the manuscript. In this case it is recommended that Sb4 be removed from the paper entirely.

Alternatively, if this foregoing suggestion is rejected, the following changes are needed.

The process used to assign backbone shifts for Sb4 should be detailed in the results and methods sections.

HSQC chemical shift perturbation/similarity analysis should be presented in new figures/panels illustrating the clear similarity for one set of Sb4 peaks to the Sb3 spectrum and the clear similarity of the other set of Sb4 peaks to the B4 spectrum.

Line 254-258 – HSQC overlays are useful but can also obscure important features that need to be readily apparent. Fig. S12 should be revised to include side-by-side comparisons of the Sb4, B4 and Sb3 spectra in addition to the overlays. This should show that the Sb4 spectrum contains roughly twice as many signals as the other two, and that patterns present in the very different B4 and Sb3 spectra are both present in the Sb4 spectrum at equal proportions – this aspect cannot be adequately judged in Fig S12 due both because the black contours are covered by red and green contours and because the images are of very low quality. In contrast, the similarity between HSQC spectra of Sb5 and B4 (Fig S13) is readily apparent, despite the poor quality of the Sb5 spectrum, and the conclusion that Sb5 populates the B fold is adequately supported.

Line 299-301 – The Sb4 unfolding curve is indeed very similar to that of Sb3, as noted below on line 309. However, Table S1 still reports $\Delta G_{\text{folding}} = 0$ for Sb4, which is quite different from Sb3 (-3.5 kcal/mol) and appears entirely inconsistent with the cooperative unfolding transition shown in Fig. 5. Table S1 should be updated with the correct $\Delta G_{\text{folding}}$ value for Sb4 and this value should be

noted in the text on line 309.

Line 309 – “Sb4 has a temperature unfolding profile very similar to Sb3 even though both S- and B- are equally populated at 25 °C in Sb4 (Fig. 5).” Fig 5 should be called out just after ‘very similar to Sb3’ since it contains the CD melts. The amended Fig S12 should be called out at the end of the sentence, since the HSQC spectrum of Sb4 is the only data in the paper supporting the assertion that “both S- and B- are equally populated”

REVIEWER COMMENTS

Reviewer #1 (Remarks to the Author):

The paper revised properly according the reviewers' suggestions. I am satisfied with the revision and do not have more concerns except the abstract may improve to be more friendly to broad audience of the journal. For example, the first sentence of "Protein sequences encoding three common small folds (3α , β -grasp, and α/β -plait) were connected in a network with high-identity intersections, termed nodes." I can understand the authors want to refer to their Fig 1 but as the first sentence in the abstract, I would like authors find a more audience-friendly way to explain their concept such as network and high-identity intersections. Does "high identity intersection" refer to "high sequence identity"? But this is more about personal writing style and my suggestion may be quite personal too. Thus, whether the language should be revised would be better decided by the editors who understand the journal audience and target better.

Response:

The abstract (lines 30-32) have been revised as follows:

"Protein sequences encoding three common small folds (3α , β -grasp, and α/β -plait) were connected in a network of mutational pathways that intersect at high-identity sequences, termed nodes."

Reviewer #2 (Remarks to the Author):

This revised manuscript is improved and addresses many of the identified weaknesses in the original, but significant concerns detailed below must be addressed before publication.

It is possible that Sb4 is exchanging between the B and S folds; the ZZ-exchange peaks provided for inspection (but not for publication) indicate that some slow conformational change is occurring. However, it is not clear how the assignment of those exchanging peaks were determined to be Gly 41 or which one corresponds to the S and B fold, since the overlay in Fig S12 shows no nearby peaks in the B4 and Sb3 spectra. From the absence of any description of the Sb4 NMR assignments, it appears that they have not been determined unambiguously. If so, the conclusion that Sb4 exchanges between the B and S folds (e.g. Fig. 8 and lines 395-6) is not adequately supported by the results presented in the manuscript. In this case it is recommended that Sb4 be removed from the paper entirely.

Alternatively, if this foregoing suggestion is rejected, the following changes are needed.

The process used to assign backbone shifts for Sb4 should be detailed in the results and methods sections.

HSQC chemical shift perturbation/similarity analysis should be presented in new figures/panels illustrating the clear similarity for one set of Sb4 peaks to the Sb3 spectrum and the clear similarity of the other set of Sb4 peaks to the B4 spectrum.

Line 254-258 – HSQC overlays are useful but can also obscure important features that need to be readily apparent. Fig. S12 should be revised to include side-by-side comparisons of the Sb4, B4 and Sb3 spectra in addition to the overlays. This should show that the Sb4 spectrum contains roughly twice as many signals as the other two, and that patterns present in the very different B4 and Sb3 spectra are both present in the Sb4 spectrum at equal proportions – this aspect cannot be adequately judged in Fig S12 due both because the black contours are covered by red and green contours and

because the images are of very low quality. In contrast, the similarity between HSQC spectra of Sb5 and B4 (Fig S13) is readily apparent, despite the poor quality of the Sb5 spectrum, and the conclusion that Sb5 populates the B fold is adequately supported.

Response:

We agree with the reviewer's comments and have provided more detail regarding the assignment of S_{b4} as follows.

In the results section, the following text has been added, with a new **Figure 7** (showing the requested side-by-side comparison of spectra, and describing the NMR/BMRB assignment and characterization of S_{b4}) and a new **Figure S12** (quantifying the amide chemical shift differences between the B-state of S_{b4} and B₄, and the S-state of S_{b4} and S_{b3}, as requested). The ZZ-exchange data showing the slow conformational exchange between the S- and B-folds of S_{b4} is now included.

Lines 258-277 "For S_{b4}, the HSQC spectrum exhibited approximately twice the number of amide cross-peaks relative to S_{b3} (Fig. 7A), suggesting that S- and B-states were populated simultaneously. This was confirmed by NMR assignment and also comparison of the HSQC spectra for S_{b4}, B₄, and S_{b3}. A significant fraction of the S_{b4} backbone amide signals (~50 peaks) closely matched those of B₄, indicating the presence of a B-state (Fig. S12A-C). The close matching of these peaks is presumably because residues 1-56 in the B-state of S_{b4} are identical in sequence to B₄. The largest amide shift perturbations between the B-state of S_{b4} and B₄ occur for residues proximal to the C-terminus of the B-fold, such as G41, where S_{b4} has additional residues and B₄ does not. Many of the S_{b4} signals also matched well with S_{b3}, although the degree of similarity was not as extensive as with B₄ (Fig. S12D-F). More significant amide chemical shift differences between the S-state of S_{b4} and S_{b3} are likely due to the Y5L mutation, which is a relatively large change located adjacent to the core. To resolve these ambiguities, backbone resonance assignments were made for the S-state of S_{b4} (Fig. 7A, BMRB 51719; see Methods for details). Comparison of S_{b4} S-state assignments with S_{b3} indicated that most of the larger amide shift perturbations were in the β 1 and β 4 strands. Secondary shift analysis showed that the pattern of secondary structure elements for the S-state of S_{b4} is similar to that of S_{b3} (Fig. 7B). Inter-proton NOE analysis indicated that the arrangement of the β -strands is also similar (Fig. 7C). Together, these results show that S_{b4} populates both S- and B-folds approximately equally at 25°C. Moreover, a ZZ-exchange spectrum demonstrated that the S- and B-states of S_{b4} are in slow conformational exchange on the NMR timescale (Fig. 7D)."

In the methods section, the following text has been added:

Lines 511-517 "The backbone resonances for the S-state of S_{b4} were assigned using triple resonance methods as above, under conditions where the S-state is more favorably populated (30°C, 100 mM KPi, 200 mM sodium chloride, pH 7.0). Amide assignments were then transferred to the two-dimensional ¹H-¹⁵N HSQC spectrum of S_{b4} at 25°C in 100 mM KPi, pH 7.0. Inter-proton NOEs for the S-state of S_{b4} were obtained at the 30°C/high salt condition, employing a 3D ¹⁵N-edited NOESY spectrum with a 150 ms mixing time. A two-dimensional ZZ-exchange ¹H-¹⁵N HSQC spectrum was recorded on S_{b4} using a mixing time of 300 ms (25°C, 100 mM KPi, pH 7.0)^{61,62}."

Line 299-301 – The Sb4 unfolding curve is indeed very similar to that of Sb3, as noted below on line 309. However, Table S1 still reports deltaGfolding = 0 for Sb4, which is quite different from Sb3 (-3.5 kcal/mol) and appears entirely inconsistent with the cooperative unfolding transition shown in Fig. 5. Table S1 should be updated with the correct deltaGfolding value for Sb4 and this value should be noted in the text on line 309.

Response:

We appreciated the reviewer finding this error. Table S1 has been corrected.

Line 309 – “Sb4 has a temperature unfolding profile very similar to Sb3 even though both S- and B- are equally populated at 25 °C in Sb4 (Fig. 5).” Fig 5 should be called out just after ‘very similar to Sb3’ since it contains the CD melts. The amended Fig S12 should be called out at the end of the sentence, since the HSQC spectrum of Sb4 is the only data in the paper supporting the assertion that “both S- and B- are equally populated”

Response:

The sentence has been revised as suggested (lines 326-327):

*“S_{b4} has a temperature unfolding profile very similar to S_{b3} (**Fig. 5**) even though both S- and B-folds are approximately equally populated at 25 °C in S_{b4} (**Fig. 7**).”*

As described in the response above, Figure 7 is a new figure showing that S- and B-folds are approximately equally populated.